# A Microservice and Serverless Architecture for Secure IoT System

**DOI:** 10.3390/s23104868

**Published:** 2023-05-18

**Authors:** Ruiqi Ouyang, Jie Wang, Hefeng Xu, Shixiong Chen, Xuanrui Xiong, Amr Tolba, Xingguo Zhang

**Affiliations:** 1School of Communication and Information Engineering, Chongqing University of Posts and Telecommunications, Chongqing 400065, China; s210131183@stu.cqupt.edu.cn (R.O.); s210131230@stu.cqupt.edu.cn (J.W.); 2Amazon.com, Inc., Seattle, WA 98109, USA; hefengx@amazon.com; 3School of Software, Dalian University of Technology, Dalian 116024, China; csxwant@mail.dlut.edu.cn; 4Department of Computer Science, Community College, King Saud University, Riyadh 11437, Saudi Arabia; atolba@ksu.edu.sa; 5Department of Mechanical Systems Engineering, Tokyo University of Agriculture and Technology, Nakacho Koganei, Tokyo 184-8588, Japan; xgzhang@go.tuat.ac.jp

**Keywords:** IoT security, edge computing, microservice, serverless architecture

## Abstract

In cross-border transactions, the transmission and processing of logistics information directly affect the trading experience and efficiency. The use of Internet of Things (IoT) technology can make this process more intelligent, efficient, and secure. However, most traditional IoT logistics systems are provided by a single logistics company. These independent systems need to withstand high computing loads and network bandwidth when processing large-scale data. Additionally, due to the complex network environment of cross-border transactions, the platform’s information security and system security are difficult to guarantee. To address these challenges, this paper designs and implements an intelligent cross-border logistics system platform that combines serverless architecture and microservice technology. This system can uniformly distribute the services of all logistics companies and divide microservices based on actual business needs. It also studies and designs corresponding Application Programming Interface (API) gateways to solve the interface exposure problem of microservices, thereby ensuring the system’s security. Furthermore, asymmetric encryption technology is used in the serverless architecture to ensure the security of cross-border logistics data. The experiments show that this research solution validates the advantages of combining serverless architecture and microservices, which can significantly reduce the operating costs and system complexity of the platform in cross-border logistics scenarios. It allows for resource expansion and billing based on application program requirements at runtime. The platform can effectively improve the security of cross-border logistics service processes and meet cross-border transaction needs in terms of data security, throughput, and latency.

## 1. Introduction

As the global economy advances and globalization intensifies, cross-border trade has become an indispensable component of the contemporary economy. Logistics, playing a vital role in facilitating cross-border trade, is undergoing constant evolution and innovation, propelling the advancement of cross-border logistics intelligence and digitalization. The advent of Internet of Things (IoT) technology has led to the proliferation of interconnected devices and systems, encompassing a wide range of sensors, smart devices, and logistics management systems engaged in cross-border logistics operations. The substantial volume of data generated by these devices and systems enables efficient interaction and information exchange via the Internet, significantly bolstering the progress of cross-border logistics intelligence.

With the pervasive integration of IoT technology, the matter of IoT security has gained heightened prominence. The interconnectedness of devices and systems within the IoT necessitates internet-based connections, whereby requests from distributed devices are centrally processed and responded to. However, this centralized approach presents inherent security vulnerabilities, including network attacks, data leakage, privacy infringements, and system security susceptibilities. Notably, within the domain of cross-border logistics, the imperative for information security and protection becomes even more critical and intricate due to the involvement of international trade, customs, and transportation factors.

The decentralized characteristics of IoT scenarios harmonize effectively with distributed application architectures. Two prominent software architecture design paradigms, namely microservice architecture and serverless architecture, have gained widespread adoption in diverse domains as integral components of distributed application architecture. Microservice architecture facilitates the construction of highly flexible and scalable systems, surpassing intricate centralized processing methodologies in terms of scalability, maintainability, and extensibility. Consequently, it offers superior adaptability to evolving business requirements, enhancing the system reliability and performance. On the other hand, serverless architecture presents a more agile and versatile development approach that expedites the deployment of novel applications while mitigating development and operational costs.

### 1.1. Motivation

Microservice architecture and serverless architecture encounter certain challenges pertaining to system integrity and information security, including the risks of information privacy breaches, interface exposure, and inadequate security and privacy measures across various microservices and functions. This paper endeavors to investigate the design framework of a cross-border logistics compliance platform founded on the principles of microservice and serverless architecture. Specifically, this study aims to explore the amalgamation of microservice and serverless architecture in augmenting the security of cross-border logistics compliance transaction platforms, while encompassing the platform’s design, development, and testing phases. The research undertaken in this paper will contribute to enhancing the stability and security of IoT applications, thereby serving as a valuable reference for related studies.

### 1.2. Research Challenge

The advent of serverless architecture, as a novel software deployment pattern, brings forth notable advantages such as maintenance-free operations and pay-per-use cost models. Meanwhile, microservice architecture, distinct from traditional monolithic software architecture, has emerged to address the requirements of contemporary internet back-end services, encompassing high-concurrency, high-performance, and high-availability aspects [1]. It holds substantial economic value [2].

While microservice architecture and serverless architecture have gained popularity as distributed software architecture design paradigms [3], challenges and difficulties persist when employing them within the context of cross-border logistics IoT security. One significant challenge revolves around ensuring the security and privacy of communication between microservices and functions. Given the independent nature of each service within a microservice architecture, disparate security requirements and privacy restrictions may apply to different services. In this regard, establishing reasonable access controls and data permissions for each microservice and function is paramount to prevent unauthorized access and information leakage. Furthermore, services and functions in microservice architecture and serverless architecture are susceptible to diverse security threats, including denial-of-service attacks and information tampering attacks. In the domain of cross-border logistics, these attacks can have detrimental consequences such as information loss.

Another critical challenge lies in ensuring the comprehensive security of microservices and serverless architectures. Within both microservices and serverless architectures, systems commonly comprise multiple services or functions dispersed across diverse locations, giving rise to intricate interactions. These interactions have the potential to introduce security vulnerabilities. Consequently, guaranteeing the continuous and seamless operation of a logistics business while upholding the security, integrity, and confidentiality of data presents a considerable challenge.

Hence, it is worthwhile to engage in further deliberation concerning the effective integration of microservice and serverless architectures, the formulation of microservice partitioning strategies, and the delineation of communication methodologies suitable for cross-border logistics platforms. Additionally, exploring secure approaches for transmitting logistics information within the framework of microservice and serverless architectures deserves considerable attention.

### 1.3. Contributions

We present the partitioning of a cross-border logistics compliance platform into microservices. The platform is separated into five distinct business domains, aiming to isolate the impact of attacks within specific service scopes. The proposed approach in this study tightly integrates the cross-border logistics compliance assessment domain with platform microservice technologies, effectively mitigating single points of failure and enhancing the overall security of the system, thus making the system more robust and maintainable.

Furthermore, we present the specific design of an API gateway and asymmetric encryption method for a cross-border logistics compliance platform. The API gateway serves as a unified interface request and permission filtering mechanism, facilitating communication between microservices and clients, thereby ensuring secure communication within the platform. The adoption of asymmetric encryption effectively encrypts communication fields, thereby reducing the risk of data leakage during transmission. The proposed design in this paper effectively addresses the vulnerability of exposed microservice interfaces to attacks, while improving the security and reliability of the platform. The primary contributions of this paper can be summarized as follows:This paper proposes a comprehensive system design that integrates the serverless and microservice architecture paradigms, with a specific focus on the context of cross-border logistics. The developed system provides robust evidence of its practical advantages;This paper presents the specific design of an API gateway and asymmetric encryption method for a cross-border logistics compliance platform, aiming to enhance security of the platform;This paper evaluates the architectural design patterns of serverless and microservice, analyzing their benefits in terms of system security, resource utilization, throughput, and latency.

The paper is organized as follows. Section 2 provides an overview of the relevant literature. In Section 3, the domain-driven design approach is employed to identify and partition the microservices utilized in cross-border logistics. Building upon the microservice division outlined in Section 3, Section 4 presents a comprehensive description of the key platform components, encompassing the overall architecture, microservice communication, and system storage. Subsequently, the platform is implemented based on these design specifications. Section 5 entails the utilization of testing tools to conduct rigorous system testing and subsequent discussions. Lastly, the paper concludes with a summary and final remarks in the final section.

## 2. Related Work

As global trade continues to advance, cross-border logistics has emerged as a vital component of international trade. However, conventional cross-border logistics models encounter challenges such as information opacity, uncontrolled logistics chains, and elevated risks. The escalating demand for global logistics information dissemination amplifies the strain on traditional systems, leading to inadequate computing resources and communication overload [4]. In this context, the advent of IoT technology presents both opportunities and challenges for cross-border logistics [5]. In recent years, cross-border logistics transportation, as a significant facet of IoT applications, has become intricately intertwined with the progression of IoT technology. As IoT technology continuously evolves and finds broader application, security and resource allocation concerns within the IoT domain have increasingly garnered attention. Similar issues, such as security and resource allocation, have been key factors constraining the development of other IoT application scenarios including smart homes, smart healthcare [6,7,8], and smart transportation. In order to ensure robust application security and optimal resource utilization within the IoT landscape, scholars and researchers worldwide have undertaken extensive research efforts, persistently exploring novel technologies and methodologies pertaining to IoT security and resource allocation.

### 2.1. IoT Security

Research on the security architecture of IoT constitutes a crucial domain within the broader field of IoT security. The security architecture of IoT must align with the characteristics and requirements unique to the IoT landscape while ensuring its integrity and trustworthiness [9]. In recent years, scholars have put forth various models for the security architecture of IoT, including the collaboration and identity-based security architecture model [10,11] as well as the blockchain-based security architecture model [12].

IoT security protocols serve as the fundamental framework for safeguarding the security of IoT communications [13]. In recent years, scholars have presented a range of IoT security protocols, including cryptographic-based security protocols [14], identity-based security protocols [15], and LoRa-based protocols [16]. These protocols aim to achieve objectives such as data confidentiality, integrity, and availability within IoT application scenarios, thereby enhancing the overall security of IoT communications.

The IoT security management platform [17] plays a pivotal role in ensuring the security of IoT applications. It facilitates comprehensive management, monitoring, and control of IoT applications, offering unified security management for IoT devices and systems. In recent years, researchers have put forth various models for IoT security management platforms, including the security management platform model based on software-defined networks [18], the security management platform model based on edge computing [19], and the IoT big data security management platform model integrating cloud computing and edge computing. These platforms aid in swiftly identifying IoT security issues and enhancing the overall security of IoT applications [20].

### 2.2. Resource Allocation in the IoT

With the widespread adoption of 5G communication technology, the IoT has experienced significant advancements. The proliferation of mobile devices in the network has led to an increased demand for real-time services [21], thereby creating opportunities for the comprehensive utilization of emerging architectural deployment models in the IoT domain. Consequently, effective resource allocation in the 5G and 6G era has become crucial, given the exponential growth of data across diverse industries, which necessitates intelligent solutions to enhance network performance and deliver high-quality services to users [22]. Dynamic resource allocation [23] is a vital part of this effort. For example, in 5G slicing, operators need to create different network environments for different users, provide different applications and services, and make real-time adjustments based on network load and user demand [24]. Moreover, to optimize operator profits, intelligent allocation and scheduling of distributed computing resources and edge computing resources can be achieved through the implementation of traffic control systems, dynamic orchestration of edge computing [25], and content caching [26]. Different IoT application scenarios demand various system architecture approaches to meet the communication, computing, and storage resource requirements.

In the upcoming 6G era, computing power will be pervasive throughout the network. The advancement of cloud computing and edge computing has facilitated the migration of conventional applications to the cloud and edge environments, enabling them to cater to a larger user base [27]. Hong et al. [28] proposed a fog computing ecosystem that extends cloud computing to terminal devices and implemented a real testbed, which was evaluated through various use cases. To address challenges related to high throughput, high latency, and limited computing resources, Ning et al. [29] proposed the integration of cloud computing and mobile edge computing to leverage the respective strengths of both approaches. David [30] introduced a hybrid cloud model that combines the economic efficiency of public cloud computing with the security and control of private cloud computing to serve private and public spaces. In order to tackle issues such as network cloud overload operations and network congestion, Zhang et al. [31,32] introduced an edge server at the network edge and designed a deep reinforcement learning-assisted federated learning algorithm to manage data transmission. By offloading a substantial number of cloud computing tasks to the edge server [33], the burden on the network cloud can be significantly reduced, thereby accelerating data processing speed. Considering the existing resources of cloud computing, fog computing, and edge computing, it is essential to flexibly utilize these resources and optimize resource scheduling [34] to meet the demands of multi-level deployment and flexible scheduling of computing, storage, and network resources for future 6G services [35,36]. Wang [37], Ning [38], and other researchers have addressed the increasing demands for connectivity and ultra-low latency by deploying fog computing in distributed traffic management systems and vehicle networks. This approach aims to alleviate the load on centralized computing centers and minimize the response time for vehicles to collect and report incidents. These advancements provide a novel perspective on IoT application architecture, wherein Mobile Network Operators (MNOs) allocate computing and caching resources to mobile users through the deployment of central control systems within the traditional network application framework [39].

In conclusion, the future application necessitates a distributed, flexibly configurable, and dynamically scalable network application architecture. This architecture is essential to cater to the ever-evolving demands of the expanding IoT application system, while enhancing the flexibility of scheduling and scalability.

### 2.3. Distributed Architecture

With the rapid advancement of IoT technology, the number of users and their demands are on the rise. Consequently, software complexity and scale have also increased. The traditional monolithic application architecture, which integrates all software modules into a single application, poses challenges in terms of development, maintenance, and task offloading [40]. This necessitates higher requirements for software modularity and scalability. To address these demands, the application service architecture has evolved from the initial monolithic architecture to the Service-Oriented Architecture (SOA) [41], and subsequently to the microservices architecture, which aligns better with the requirements of the Internet.

The concept of microservices architecture was first introduced by Martin Fowler and James Lewis in 2014 [42]. In recent years, with the rapid growth of the mobile Internet, the monolithic application architecture has become inadequate to meet the requirements, leading to the widespread adoption of microservices architecture in various industries. Aligned with the application concepts of distributed architecture [43] and virtual network mapping [44], the microservices architecture encapsulates relatively independent applications into different services, isolates the business logic, and deploys each service separately on different servers. Container management is used to control each service, ensuring efficient management and deployment of the system components.

To alleviate developers from the burdensome tasks of server management, serverless technology introduces the concept of cloud services into the computing model [45,46]. This model effectively separates application developers from servers, relieving them of the responsibilities associated with server management and security. Additionally, cloud service providers host the underlying infrastructure, eliminating the impact of device differences on upper-layer applications. Compared to traditional computing models, serverless technology exhibits excellent performance in terms of high concurrency, low latency, and other aspects [47]. Consequently, scheduling and resource management [48,49] functions, as well as distributed application practices [47], are vital areas of research in the field of serverless technology. In the serverless model, computing resources are billed based on execution time, with billing stopping immediately after request processing is completed. This computing model proves to be cost-effective for applications with varying levels of business requests.

In this section, we provide a concise overview of three key aspects: current IoT security solutions, IoT resource allocation solutions, and the existing architecture of distributed systems (Table 1). In this paper, we consider these three aspects collectively to inform the design of our platform.

## 3. System Design

In this section, we will perform requirements analysis and preliminary preparations based on the application scenario. Unlike the requirement phase in traditional waterfall development, this paper adopts the DDD method to establish the consistency between business and code logic through the abstraction of the business and the establishment of the domain model. This approach ensures that the business requirements are effectively translated into the software design and implementation process, enabling a more robust and aligned solution.

### 3.1. Domain-Driven Design

DDD is a methodology that helps to mitigate the confusion between the complexity of business logic and technical implementation. It achieves this by establishing clear boundaries between the business logic and technical aspects, effectively isolating their respective complexities. By doing so, DDD ensures that the business rules remain unchanged regardless of the underlying technology employed. The ultimate goal is to maintain an orthogonal relationship between business logic and technical implementation, where changes in technology do not impact the core business rules. This approach enables greater flexibility and adaptability in the system’s design and evolution.

The process of applying Domain-Driven Design (DDD) involves several essential steps. Firstly, the problem domain is identified by clarifying the business context and user vision, which helps establish a shared understanding and language with domain experts, thereby laying the foundation for subsequent domain modeling activities. Secondly, the major business processes are identified through techniques such as user story mapping and event storms, enabling the mapping of user interactions and determining the pivotal events in the system. Subsequently, closely related domain models are aggregated by identifying associations and relationships among them. The overall domain model for the entire system is then determined by employing bounded contexts as the boundary for microservice partitioning. Throughout the entire DDD process, the domain models and technology models are tightly integrated to ensure their clarity, completeness, and strong consistency throughout the software development lifecycle [50]. These inherent characteristics of DDD align well with the principles of microservice architecture, making DDD a key determinant of success in microservice applications [51].

#### 3.1.1. Clarify User Vision

Firstly, it is crucial to establish a clear user vision for the system. The system primarily targets three distinct user categories:Cross-border logistics customers: This user group constitutes the primary target audience for the system, as they utilize the cross-border logistics platform to facilitate the transportation of their products across international borders.Compliance assessment service providers: These users play a crucial role in the system as they offer business consultation and information services related to compliance assessment for cross-border logistics sellers.System platform administrators: While secondary users of the system, these individuals are responsible for overseeing and managing the system platform. They require access to statistical data on compliance applications within the system to enhance the logistics compliance process.

To develop the proposed system, extensive research and discussions were conducted with domain experts who have expertise in cross-border logistics business, involving customers, compliance evaluation service providers, and platform administrators. The insights gathered from these discussions were utilized to create the product vision board, as depicted in Figure 1.

The vision board above summarizes the core business requirements of the product, which are as follows:Cross-border logistics customers will have access to comprehensive compliance requirements tailored to their target markets. This will enable them to ensure compliance during transportation and conduct self-checks efficiently.Through the system platform, cross-border logistics customers can discover and connect with suitable compliance evaluation service providers who can offer support and certification services.Compliance evaluation service providers can utilize the system to acquire customers and deliver compliance certification services to them.System administrators will have the capability to create, update, and maintain the compliance evaluation rules files within the system.System administrators can manage the information of compliance service providers by adding, deleting, and modifying their details.System administrators will have access to system access information, including activity records of logistics customers and service providers, as well as product-type search records. This information can be used to enhance the system performance and effectiveness.

These functionalities are designed to streamline the cross-border logistics process, enhance compliance management, and facilitate efficient collaboration among customers, service providers, and administrators.

#### 3.1.2. Identify Problem Domain

The different stakeholders involved in a cross-border logistics platform have specific requirements and concerns. Logistics customers are primarily focused on accessing import and export compliance policies relevant to their specific products. They also seek to connect with suitable compliance service providers to obtain comprehensive compliance assessment reports. These reports are subsequently presented to the cross-border logistics platform to obtain approval for product transportation. In order to fulfill these requirements, the system needs to maintain a database that links compliance policies with corresponding products. Additionally, the system should facilitate communication between customers and service providers by notifying the latter when customers attempt to contact them. This ensures a seamless and efficient process for compliance assessment and approval within the platform.

#### 3.1.3. Sorting Out the User Story Map

After defining the system’s problem domain and business requirements, the main business processes were identified. These processes represent the collaborative activities of different stakeholders to accomplish domain functions that contribute to the business value.

User story mapping is a valuable technique (Figure 2) that aids in structuring user stories into meaningful models. It helps in comprehending the system’s functionality, identifying any gaps or missing elements in the backlog, and efficiently planning releases that deliver value to both users and the business. This approach enables a holistic understanding of user needs and ensures that development efforts align with the desired outcomes.

#### 3.1.4. Event Storm and Command Storm

The term “event” refers to an actual occurrence within the business domain. Event storming sessions facilitate collaboration among domain experts and workshop participants to clarify changes in domain objects and the corresponding attention that the software system should give to business data changes during the business process. Events possess the following characteristics: they hold significance in the business context, are expressed in the past tense, and have a temporal order. When identifying events, only the “write model” is considered, excluding the content of the “read model”. The write model encompasses changes in the state of domain objects resulting from business decisions, such as creation, update, termination, deletion, etc. The read model primarily focuses on data retrieval and presentation, without causing changes in the state of domain objects.

Based on the outcomes of the event storming session, a command storming session outlines the commands that trigger the events. Events serve as outputs within the business domain, while commands serve as inputs within the business domain. The purpose of a command storming session is to identify the business actions or decisions that generate events. This session helps identify the system’s final functionalities that will be utilized by external users. The commands and events identified during this session will guide the design of the system’s API in subsequent stages. Command descriptions should adopt verb-object phrases and align as closely as possible with the business terminology established in the common language. Commands solely concern the write model in the system and do not address the “read model”. Therefore, commands such as “query products” are not included in this context.

#### 3.1.5. Finding Aggregation

Aggregation entails gathering a set of interconnected domain models that encapsulate business invariants and promote strong cohesion among tightly-coupled models. The objective of employing aggregation is to encapsulate business invariants and encourage the simplification of associations between domain models, thereby achieving a state of high cohesion and low coupling within the business layer.

#### 3.1.6. Bounded Context

The aggregates are categorized based on their contextual significance. The User Aggregate encompasses the User Domain, which handles tasks such as user information updates, user authentication, and related functionalities. The Rule Aggregate represents the Rule Domain, responsible for ensuring compliance with product requirements and managing associated content. The Order Aggregate represents the Order Domain, handling order-related operations. The Monitoring Aggregate represents the Monitoring Domain, responsible for statistical analysis of user request data. The Common Aggregate represents the Common Domain, providing functional components that other domains may utilize. The resulting bounded context is depicted in Figure 3.

### 3.2. Microservice Splitting

Based on the outcomes of the bounded context analysis, the microservices can be partitioned. In principle, each bounded context corresponds to a single microservice; however, factors such as service responsibility and team heterogeneity must be considered during the implementation process. In this system, the results of the bounded context directly inform the division of microservices. The final configuration of microservices comprises three primary domains: user domain, order domain, and rule domain. In addition to these core domains, there is a statistics domain that facilitates administrator information retrieval, as well as a general domain that supports system operations. Further details regarding all the microservice domains in the system can be found in Table 2.

## 4. System Implementation

Based on the microservices design and partitioning, we have developed a comprehensive architecture for the platform. In this section, we present the system platform that combines microservice and serverless architectures, addressing various aspects including the overall architecture diagram, system aspect diagram, process view, microservice communication mode, and microservice interaction design. The system architecture design provides detailed insights into the overall technical architecture of the system, the layered service logic structure, and the data flow within the system. The continuous integration/continuous deployment design outlines the deployment methods for both the frontend and backend components, along with the step-by-step procedures involved. Additionally, the microservice communication mode explains the rationale behind the platform’s technology selection and proposes an API gateway solution.

### 4.1. System Architecture Design

During the architecture design process, estimating the traffic load of the system proves challenging due to the introduction of a new business scenario. Deploying applications directly on cloud servers using traditional approaches would require manual horizontal scaling to accommodate insufficient traffic, potentially resulting in resource wastage if multiple servers are provisioned in advance. To address this issue and align with business requirements, a serverless architecture based on AWS was adopted for implementation. By deploying microservices to AWS Lambda and leveraging edge computing, the application can automatically scale elastically based on the traffic volume, with costs calculated according to the number of requests and computing time.

In order to promptly respond to evolving user requirements, enhance development efficiency, and expedite system delivery, the adoption of continuous integration and continuous delivery (CI/CD) is essential to automate the entire release process. Automating tasks such as code compilation, building, and deployment minimizes the need for manual monitoring of every change and reduces system risks by proactively identifying potential issues. This automation process encompasses not only code-related activities but also the dynamic creation, destruction, and updating of resources necessary for system operation, including servers and databases.

The overall serverless architecture design of the system is shown in Figure 4.

The technical architecture of the system primarily comprises the following components:Microservice backend module composed of Lambdas.

The backend utilizes AWS Lambda as the computational unit to implement a serverless system architecture, leveraging its automatic scaling capabilities and cost-efficiency based on pay-per-use principles.

Frontend module built using React and hosted on cloud services.

The frontend functional modules are developed using React, and the compiled files are deployed to S3 and distributed through CloudFront CDN to enhance the accessibility of global users and improve performance.

Serverless general component services provided by cloud service providers.

The general serverless components encompass DynamoDB NoSQL databases, message queues, email services, and object storage services. While the database is specifically dedicated to the microservice, the remaining components are designed as general components accessible to the microservice computing module (Lambda).

Continuous integration and continuous deployment (CI/CD) pipelines.

The CI/CD pipeline is segregated into two autonomous pipelines for the frontend and backend modules. Each pipeline encompasses the code repository, build service, and deployment unit. Upon detecting modifications in the primary branch of the code repository, the pipeline initiates execution and, upon approval, deploys the changes to the designated account environment. The incorporation of CI/CD facilitates accelerated development and diminished delivery time.

Other agile facilities.

To bolster development process agility and ensure product quality, the Scrum methodology was employed in the system construction process of this project. The Asana Kanban tool was utilized to allocate tasks via story feature cards and update task statuses during daily stand-up meetings. The project was segmented into multiple sprint stages, with a comprehensive evaluation of strengths, weaknesses, and encountered challenges at the conclusion of each stage.

#### 4.1.1. System Logical View Design

The logical view of the cross-border logistics system platform is depicted in Figure 5. The platform’s fundamental structure revolves around multiple microservices that are segregated and operate independently, while also sharing essential infrastructure components. The frontend initiates requests to the relevant microservice component via the API Gateway, whereby each service responds to distinct events.

The system architecture consists of the following layers:The access layer provides functionalities such as authentication, authorization, protocol conversion, traffic restriction, and log monitoring. It ensures secure and controlled access to the system’s resources.The anti-corruption layer acts as a boundary between the new architecture and any existing legacy systems, ensuring that the new architecture is not constrained by the limitations of the old system. It facilitates seamless integration and communication between the two.The domain layer encompasses the definitions of all entities and encapsulates the core business logic of the system. It represents the heart of the application and implements the business rules and processes.The infrastructure layer consists of the communication code that interacts with various external services or middleware. It handles the integration with external systems, such as databases, third-party APIs, or messaging systems, enabling smooth data exchange and connectivity.

#### 4.1.2. System Process View Design

The process view of the architecture provides an abstract representation of how different components interact with each other. In this system, the main components involved in collaboration are the front-end RESTful requests, data persistence, message notifications, and file I/O operations (Figure 5).

As shown in Figure 6, the process view of the system is illustrated by a typical scenario where users send requests. The process begins when the user accesses the front-end page, which is obtained by accessing the CloudFront CDN. The static content of the front-end, such as HTML, CSS, and JS, is cached and stored by S3.

When a user makes an HTTP request to the backend API from the front-end page, the request is first proxied by the API Gateway. The API Gateway may also include authentication for certain APIs. The API Gateway then forwards the request to the corresponding Lambda microservice based on the URI resource.

Lambda, being a serverless computing unit, executes the business logic based on the requirements. It interacts with various resources as needed, such as DynamoDB for data persistence, SQS for message notifications, SES for email services, and other components for file I/O operations.

This process view provides an overview of how the different components collaborate to handle user requests and perform the necessary operations to fulfill the system’s functionalities.

### 4.2. Continuous Integration/Continuous Deployment Design

The current system follows a front-end/back-end separation design, which results in separate deployment pipelines for each. In the following sections, we will discuss the design of the front-end and back-end components, the communication pattern of microservices, and the deployment design of the persistence layer.

#### 4.2.1. Front-End CI/CD Streamline Design

The continuous deployment pipeline for the frontend, as shown in Figure 7, consists of the following steps:The frontend is developed using the React framework, and Jest and Enzyme are used as unit testing tools.Feature branches are used for development, and the main branch must be in a buildable state for deployment to the production environment with each version. When merging branches, developers need to submit a merge request (Pull Request) to obtain approval for the merge, which must be approved by at least two other members before merging is allowed.When the main branch is updated, the build enters the pipeline stage, and the code is built using Webpack through CodeBuild. Before building, a global test is performed. If the test fails, the build will fail. Once the build is successful, the built frontend static files are automatically stored in S3.In the deployment stage, the contents are pulled from the S3 storage bucket used for development and then S3 is used as a static web server to serve clients. The same process is used for the production environment.Cloudfront is used as a CDN and can be integrated with S3. S3 is used as the CDN source, and users only need to access the Cloudfront address.

#### 4.2.2. Back-End CI/CD Flow Line Design

The backend deployment pipeline is illustrated in Figure 8. The backend design can be divided into the following steps:

The backend design can be divided into the following steps:Adopt the API-first development approach, where the frontend and backend teams determine the data format required by the API. Once the data format is established, both teams can enter an efficient and independent development phase.During the development of the Lambda functions in the backend, a test-driven development approach is employed to reduce unnecessary bugs after deployment, thus increasing the system’s stability.Feature branching is used as a way of developing, where the main branch must be in a buildable state and deployable to the production environment for each version. When merging branches, developers need to submit a merge request (Pull Request) to obtain approval, which must be approved by at least two other members before the merge can be carried out.When the main branch is updated, the build process will enter the pipeline phase. The code will be built using CodeBuild, and a global test will be performed before the build. When the test fails, the build will be flagged as failed. After a successful build, the backend code will be packaged and uploaded to S3.During deployment, resources required for the backend service, including computing resources Lambda, application entry API Gateway, and data storage DynamoDB, will be generated based on the CloudFormation built.After verification of the testing environment, the deployment to the production environment can be approved.

The above describes the continuous integration/continuous deployment architecture of the system, where the continuous deployment pipeline can automate the entire software release process, thereby accelerating the delivery speed.

### 4.3. Microservice Communication Model

APIs play a crucial role in software development. In monolithic applications, APIs are commonly defined using programming language constructs, where the implementation details of specific classes are hidden from clients. In contrast, microservice architectures involve services running as separate processes on different machines, necessitating inter-process communication (IPC) for interaction. Consequently, IPC holds greater significance in microservice architectures compared to monolithic applications. Consequently, developers working on microservice applications invest more time in designing and considering communication patterns between services.

In monolithic applications, module interfaces are typically defined using programming language constructs, shielding the specific implementation classes from clients. However, in statically typed compiled languages, if an interface becomes incompatible with clients, it will lead to build failures. Unlike programming language constructs, microservice design does not offer a standardized method for constructing APIs. Clients and servers are not compiled together, and incompatible APIs can result in runtime failures.

#### 4.3.1. API Gateways

As depicted in Figure 9, within a conventional application architecture, clients typically interact directly with the application’s APIs. However, in the context of microservices, exposing all service APIs directly can give rise to the following issues:Different client types may have varying requirements for the API responses. For instance, mobile clients may require less data than web clients.Some application services may use other communication protocols such as gRPC, which are easier to adapt to service-to-service communication but difficult to adapt for mobile clients.Adding an authentication and authorization module to each microservice can impact the system’s stability and increase coupling. Any modifications require modifications to all services.Once an API is determined, modifying it becomes difficult. When backend developers want to break down services, it is difficult to update the clients, making it hard to modify the API.

To mitigate the aforementioned concerns, the system design incorporates an API gateway to encapsulate the microservice APIs [52]. This gateway intercepts all incoming request data and subsequently forwards the requests to the appropriate backend services. The architectural depiction of the API gateway can be observed in Figure 10, and its functionalities are described as follows:

Request Routing: all client requests first arrive at the API gateway, which queries the route mapping and forwards the request to the corresponding backend service. This function serves as a reverse proxy for backend microservices.Protocol Conversion: client requests are often in the form of HTTP-based RESTful requests, while backend services may use gRPC. In this case, the API gateway can perform protocol conversion, reducing client implementation costs.Authentication and Authorization: the API gateway determines client access permissions by verifying the client request identity.Speed Limit: limits the number of client requests per second, reducing system pressure.Log Monitoring: important API requests can be logged and monitored to further enhance system security.

By leveraging the API gateway, clients are relieved of the burden of handling routing, protocols, and other complexities, thereby reducing the implementation challenges on the client side. Additionally, backend services can seamlessly integrate features such as authentication, authorization, rate limiting, and monitoring, thereby reducing system coupling and enhancing system security.

#### 4.3.2. Microservice Interaction Methods

In a monolithic application architecture, communication between modules is typically facilitated through programming language-level methods or functions. However, in a microservice architecture, where each service instance operates as an independent process, inter-process communication becomes essential. Consequently, when designing a system architecture, careful consideration must be given to the communication patterns between services, ensuring that the communication modes encompass one-to-one, one-to-many, synchronous, and asynchronous interactions (Table 3).

In the system design, synchronous communication is directly implemented through HTTP requests to the application programming interfaces (APIs), while asynchronous communication primarily relies on two serverless services: SNS (Simple Notification Service) and SQS (Simple Queue Service). SNS serves as a publish–subscribe messaging service, whereas SQS functions as a message queue service that supports a producer–consumer model. Each service is deployed independently and accompanied by its respective database. To ensure efficiency and data consistency, cross-interaction between databases associated with different microservices is established [53]. By utilizing Amazon SNS and Amazon SQS together, a message can be simultaneously delivered to multiple consumers. Figure 11 illustrates the integration design of Amazon SNS and Amazon SQS.

### 4.4. Cloud Resource Management and Continuous Deployment

To streamline the management of cloud computing resources, mitigate the uncertainty associated with manual configuration, and ensure consistency between testing and production environments, CloudFormation is employed. It allows for the management of various cloud computing resources such as Lambda, DynamoDB database, API Gateway, and SQS queue by defining them in YAML files, eliminating the need for manual server resource configuration.

The entire system, including compilation, building, and deployment, is automated to minimize the workload of deploying new code. The system encompasses five backend microservices and their scheduled task Lambdas. This enables developers to rapidly release the application and receive valuable user feedback.

Given the presence of sensitive user privacy information in the database, and the inherent lack of filtering and protection mechanisms in storage devices for personal privacy data [54], ensuring the security of user information is of paramount importance and presents a significant challenge for developers. To guarantee data privacy and security, the system employs AWS KMS (Key Management Service) for key management and leverages asymmetric encryption to encrypt sensitive data fields. This approach, while providing relative resilience against cracking attempts, also safeguards against key compromise or leakage.

The process of encrypting data using KMS is illustrated in Figure 12, and can be described as follows:A request for a new data key is issued under the CMK. Encrypted and plaintext versions of the data key are returned.In the AWS Encryption SDK, the plaintext data key is used to encrypt the message. The plaintext data key is subsequently deleted from memory.The encrypted data key and the encrypted message are combined into a single ciphertext byte array.

The process of decrypting data using KMS is illustrated in Figure 13 and described as follows:Analyze the encrypted message in the envelope to obtain the encrypted data key, and request the AWS Encryption SDK to decrypt the data key.Receive the plaintext data key from the AWS Encryption SDK.Use the data key to decrypt the message and return the original plaintext.

## 5. Results and Discussions

The primary aim of this section is to assess the benefits of automatic scaling in serverless architecture by conducting load and stress testing on the system. To prevent excessive network throughput on a single server, multiple servers are utilized to concurrently generate requests. The system’s load capacity is evaluated using various metrics, such as average response time, total number of requests, successful and failed requests, and throughput rate (RPS). In addition to verifying the program’s stability and reliability, stress testing allows for an assessment of the application’s load capacity based on the test results, providing insights for potential program optimization.

### 5.1. Experimental Environment

Taurus is an open-source tool utilized for conducting a variety of load and functional tests. For this experiment, two Ubuntu 20.04 system hosts were configured, with Taurus being installed on one device to test the platform developed in this study installed on the other.

In terms of the load testing tool Taurus, there is a pre-packaged distributed load testing solution available. This solution packages Taurus into containers, facilitating easy deployment to multiple test servers. The front-end page allows for the configuration of parameters such as the number of test servers, thread numbers, target API, and duration. The entire testing process is depicted in Figure 14.

### 5.2. Test Methodology and Results

The testing conducted in this study involved two methods: load testing and stress testing.

In the load testing method, Taurus was utilized to interact with the system’s API interface. By simulating user requests, the API responses were obtained in the form of single requests. To subject the server to stress load, the number of servers and Taurus threads was increased. The distributed test output using Taurus with 100 servers and 100 threads per server is illustrated in Figure 15.

In the scenario of a large number of distributed request loads within a short period, the self-scaling system proposed in this study expanded the bandwidth resources to 1.98 Kps, with an average latency of 1.82070 s and a system response error rate of 0.0098%. These comprehensive test results demonstrate that the designed system is capable of maintaining high performance, reliability, stability, and information processing capability when confronted with a substantial number of requests.

During stress testing, tools are used to simulate a large number of concurrent users or requests, evaluating the system’s ability to handle heavy loads and maintain key performance metrics such as response time, availability, and scalability during peak loads. The testing results, as presented in Table 4, provide insights into the system’s response time and success rate metrics as the number of testing servers varies from 1 to 50. With the deployment of 100 servers, each with 100 threads, the system achieves its peak Requests Per Second (RPS) at 5425. However, the average response time of requests hovers around 1.8 s, which can be attributed to network congestion resulting from the excessive number of Taurus threads simultaneously sending requests from a single testing server.

Through CloudWatch monitoring, the growth curves of backend Lambda invocation and instance scaling are illustrated in Figure 16. As the number of requests increases, Lambda automatically scales its service instances to handle the significant volume of requests. The system proposed in this study demonstrates self-scaling of service resources based on actual request traffic and environmental demands, thereby avoiding resource shortages or waste associated with rigid, centralized software architectures.

This section aims to validate the advantage of combining serverless architecture with a microservice framework in terms of automatic scaling, which is achieved through load testing and stress testing. The test results provide evidence that the system can dynamically scale its computing resources based on the number of requests without requiring manual intervention, and the achieved throughput meets the demands of most business scenarios. Furthermore, the system design effectively mitigates the risks of single point of failure and ensures system stability and security, even in the face of attacks such as denial of service resulting from a large influx of requests.

## 6. Conclusions

This paper presents a comprehensive exploration and application of a security solution for IoT applications, leveraging the combined benefits of microservice and serverless architectures. Initially, the study investigates the architectural design patterns that integrate serverless with microservice, addressing challenges encountered in applying microservices within the serverless context, such as service decomposition and API design. Corresponding design solutions are proposed to overcome these challenges. Subsequently, an architecture centered around serverless is designed for practical business projects, accompanied by a description of the core functionalities. Performance testing is conducted on the implemented system, and the obtained results validate its effectiveness in enhancing system security and meeting the demanding requirements of cross-border transactions involving massive IoT application data, encompassing aspects such as data security, throughput, and latency.

The findings demonstrate that the proposed architecture excels in meeting the complex requirements of cross-border logistics IoT systems, mitigating the risk of single point of failure, and enhancing system availability and reliability. By employing small, independent services through microservice and serverless architecture, the system becomes less susceptible to targeted attacks, while enabling efficient security management and monitoring. Furthermore, the adoption of serverless architecture contributes to mitigating internal risks and improving overall system security. Thus, the combination of microservice and serverless architecture emerges as a feasible and promising solution for addressing security concerns in the domain of cross-border logistics IoT systems.

## Figures and Tables

**Figure 1 sensors-23-04868-f001:**
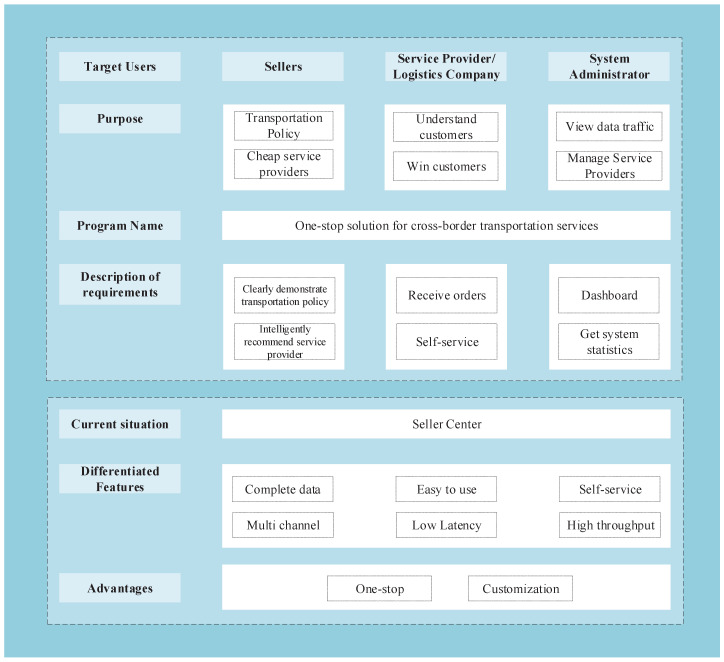
System vision board.

**Figure 2 sensors-23-04868-f002:**
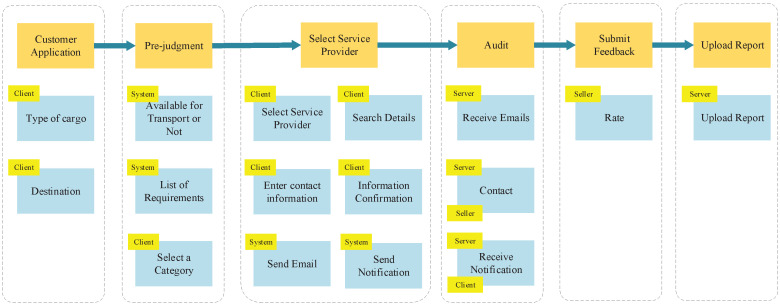
User story map.

**Figure 3 sensors-23-04868-f003:**
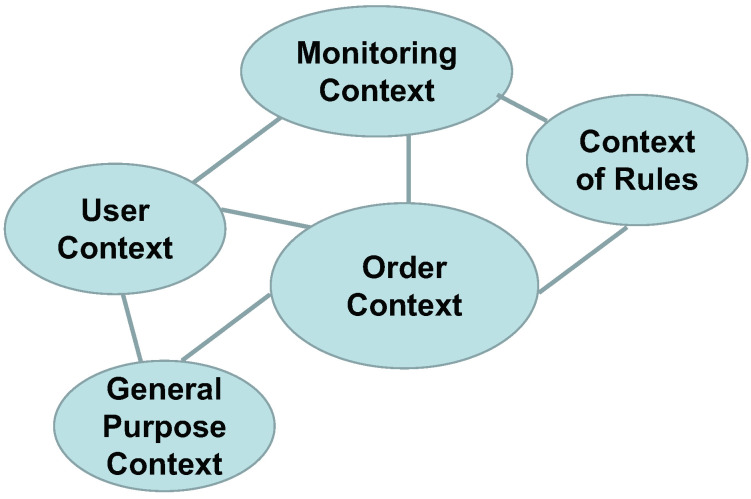
Bounded context.

**Figure 4 sensors-23-04868-f004:**
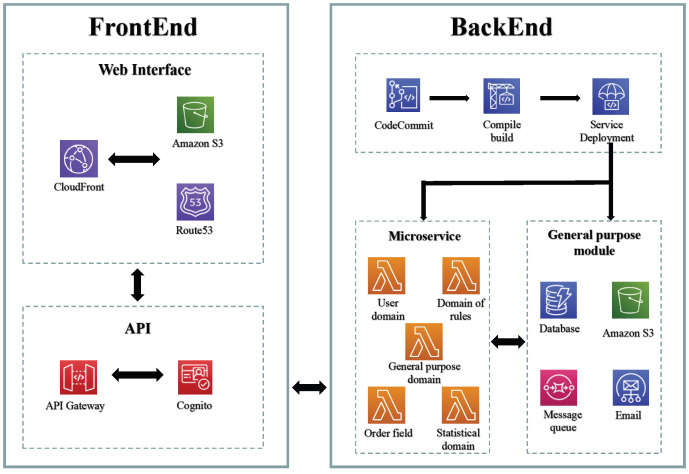
System architecture.

**Figure 5 sensors-23-04868-f005:**
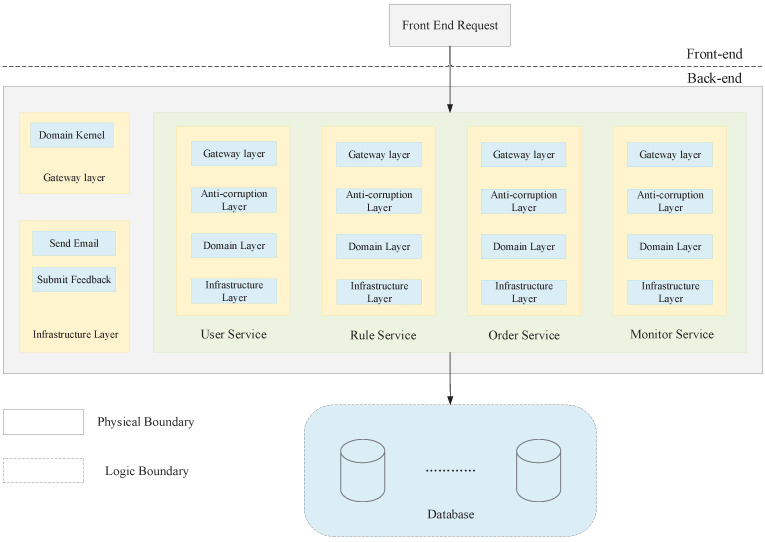
System Logic View.

**Figure 6 sensors-23-04868-f006:**
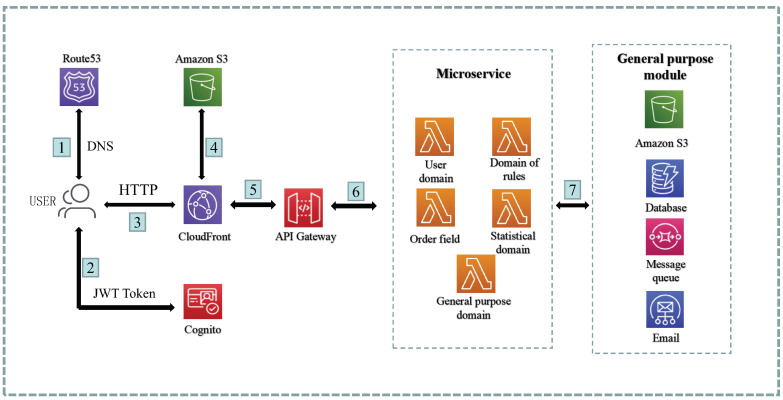
System process view.

**Figure 7 sensors-23-04868-f007:**
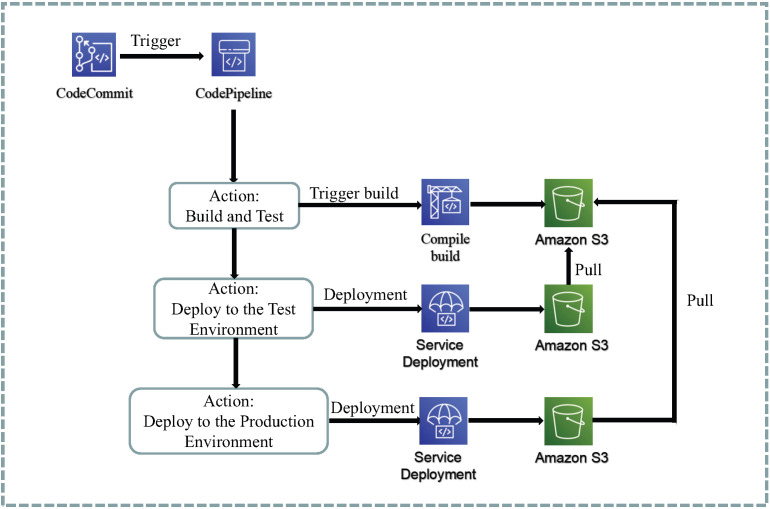
Front-end pipeline.

**Figure 8 sensors-23-04868-f008:**
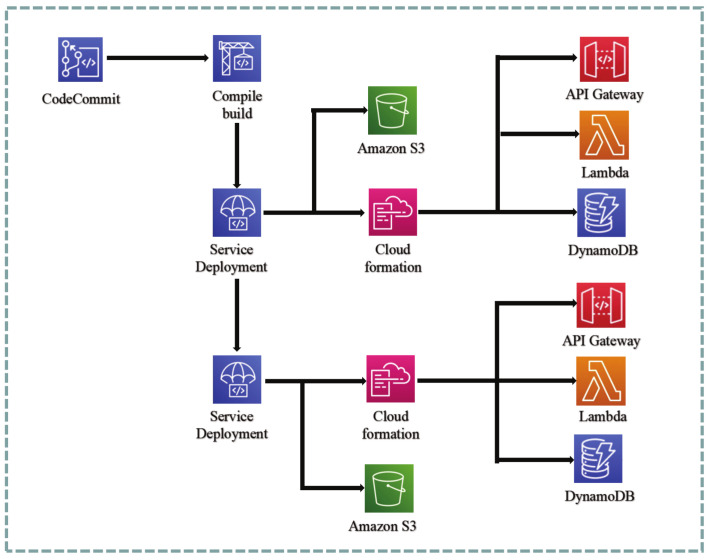
Back-end pipeline.

**Figure 9 sensors-23-04868-f009:**
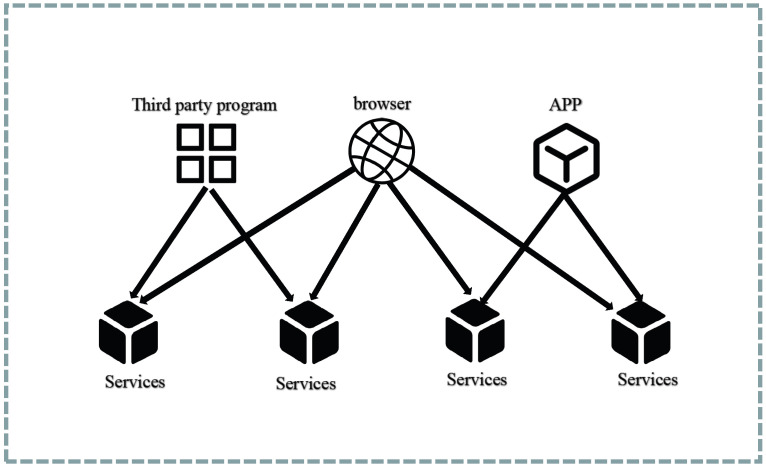
Traditional pattern.

**Figure 10 sensors-23-04868-f010:**
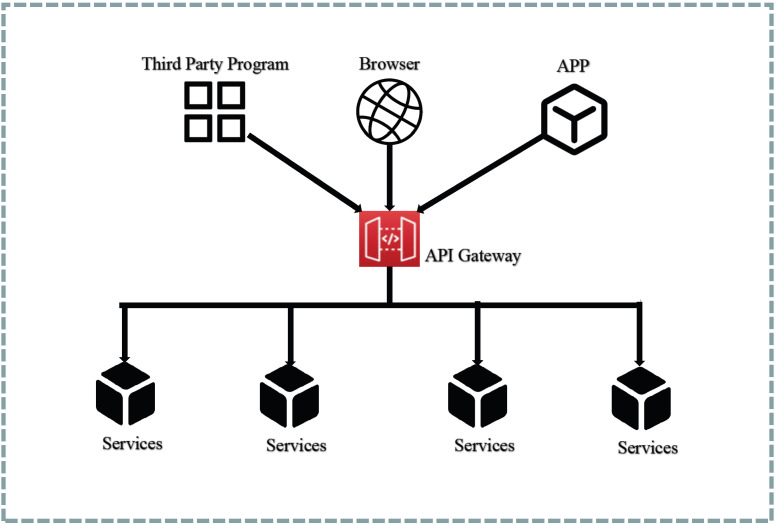
API gateway pattern.

**Figure 11 sensors-23-04868-f011:**
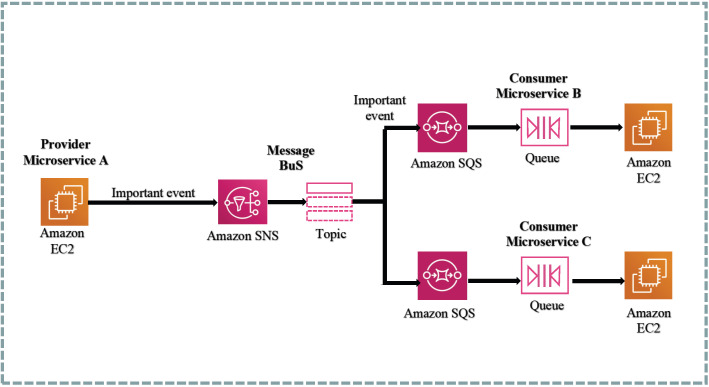
SNS And SQS example.

**Figure 12 sensors-23-04868-f012:**
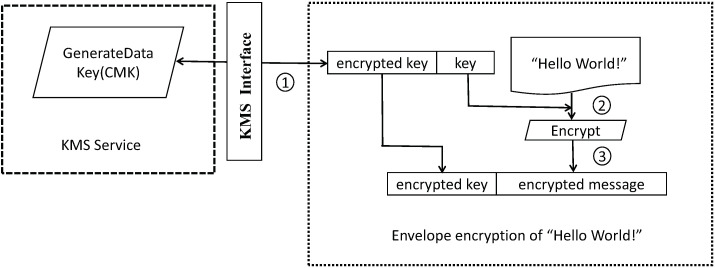
KMS encrypt.

**Figure 13 sensors-23-04868-f013:**
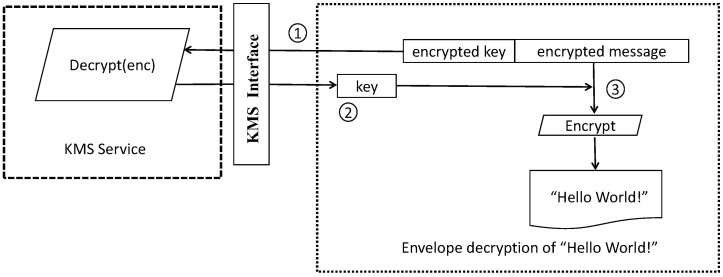
KMS decrypt.

**Figure 14 sensors-23-04868-f014:**
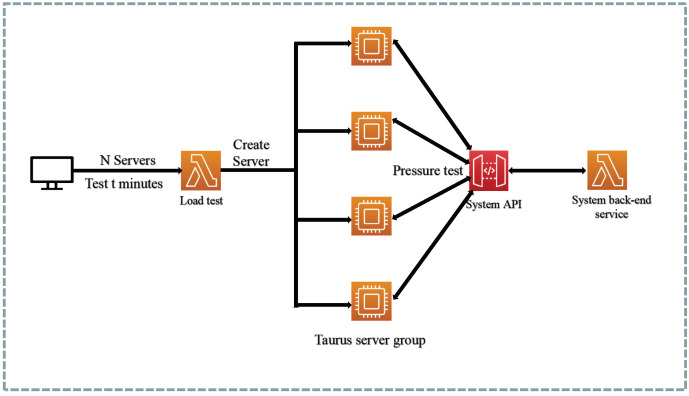
Taurus distributed load test solution.

**Figure 15 sensors-23-04868-f015:**
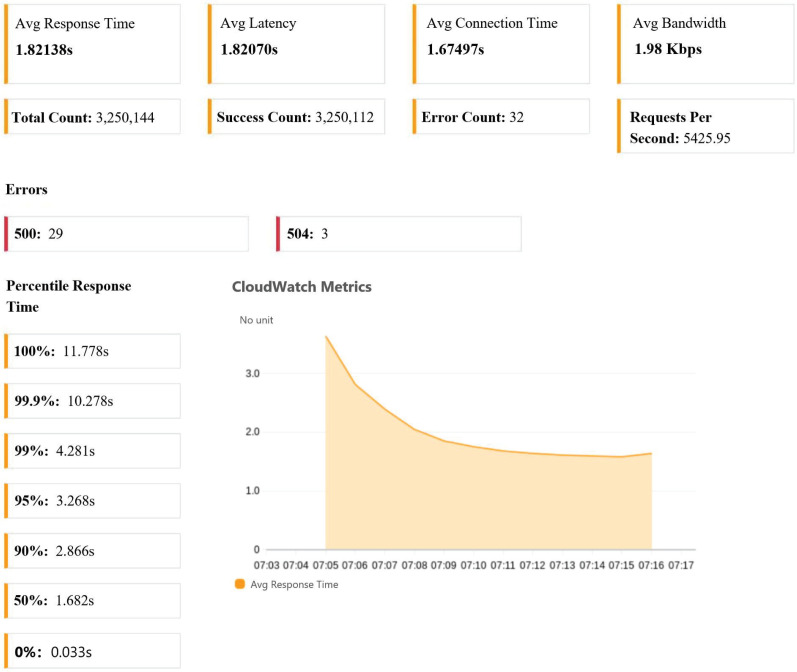
Taurus test outputs.

**Figure 16 sensors-23-04868-f016:**
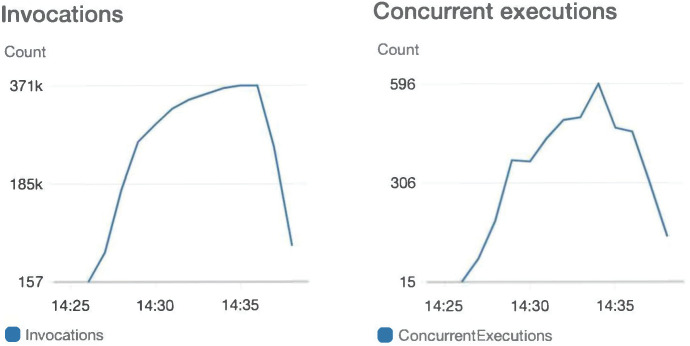
Invocations and concurrent.

**Table 1 sensors-23-04868-t001:** Summary of related work.

Purpose	References	Methods	Advantages	Disadvantages
Security of IoT	[10,11,12,14,15,16,19,20]	Blockchain, Security Protocol, Edge Computing	Improve the Security of The IoT Communication	High Computation and Communication Overhead
Resource Allocation of IoT	[25,26,27,29,31,32,37,38]	Dynamic Choreography of Edge Computing, Cloud Computing, Fog Computing	High Real-time Performance, Low Network Pressure	High Deployment Overhead, Poor Scalability
Distributed Architecture	[41,42,46]	Service-Oriented Architecture Microservice Architecture, Serverless Technology	High Development Efficiency, Lightweight and Low Cost	Maintenance Difficulty, Business Splitting and Decentralization

**Table 2 sensors-23-04868-t002:** Microservices.

Microservice Name	Function Description
User Domain	Responsible for user management, including registration, login, add, forget password, etc.
Rule Domain	Manage contents related to product compliance requirements
Order Domain	Requests sent by sellers to service providers as order types
Monitor Domain	Collection and statistics of user request access data, providing functional view for administers.
Common Domain	Services open to all domains, including mail delivery, message queue and other facilities.

**Table 3 sensors-23-04868-t003:** Microservice interactions.

Mode	One-to-One	One-to-Many
Synchronous mode	Request/Response	None
Asynchronous mode	Asynchronous Request/ Corresponding One-way Notification	Publish/Subscribe Publish/Asynchronous Response

**Table 4 sensors-23-04868-t004:** Load test.

Number of Servers	Number of Concurrent Threads	Throughput	Average Response Time	Total Number of Requests	Number of Failed Requests
1	100	49.8/s	1.97874 s	29,830	0
5	100	268.52/s	1.8458 s	160,843	0
10	100	549.16/s	1.79907 s	329,493	0
50	100	2698.09/s	1.82738 s	1,618,854	0
100	100	5425.95/s	1.82138 s	3,250,144	3

## Data Availability

No public datasets were used to support this study.

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
