# Peer review of "A Microservice and Serverless Architecture for Secure IoT System"

_sensors, 2023, doi:10.3390/s23104868_

Round 1

Reviewer 1 Report

A Microservice and Serverless Architecture for Secure IoT System

1.     The abstract is not clearly explaining the purpose of this research.

2.     The significance of the research is not mentioned properly, instead, there is only a focus on the solution.

3.     Similar sentences are used multiple times in the research paper.

4.     The significance of the proposed system and how it is making a better impact than existing systems are not defined clearly.

5.     The contribution section just explains the benefits of some components instead of explaining real contributions.

6.     There is a need to explain the existing cross-border logistics industry working process. Also, how the proposed architecture will help in meeting the desired goal.

7.     As mentioned, ensuring the security and privacy of communication between microservices and functions is a challenging process, then how it will enhance the security in the proposed solution?

8.     All the terms in related work, are defined explicitly and there is no comparison of existing systems with the proposed system.

9.     The system design is explained generically, however, it is lacking in explaining the application of system design to the proposed work.

10.  The whole paper is explaining new terms, instead of aligning the concept with the problem.   

11.  The paper needs attention on compiling the concepts in a better way, so, the reader remains concentrated on the sections that are leading towards the solution.

12.  There is a need to rephrase some statements so they depict a clear image of the proposed system.

13.  There is a need to discuss the technological advancements you have added to your research.

14.  Multiple terms are mentioned in the paper with an abbreviation, so try to mention the complete phrase.

15.  Mention at least a one-paragraph description of software or tools that are used in the research paper. As in system architecture, there are so many tools used, however, in the whole paper, not a single statement explains the purpose of using them.

16.  The result and Discussion section is so brief, as it not explaining the outcomes and evaluation parameters.

17.  The result and discussion section is not summarizing the proposed solution efficiently.

18.  The result section is not explaining the evaluation method used to find out whether the proposed system is meeting the development needs.

19.  There is a single table in the result section that is describing the load testing. There are other parameters such as data security and latency, and how it is improved is missing in the discussion.

20.  No real-life scenario or evaluation measures are used to find out the efficiency of the proposed solution.

21.  There is a need to explain how the proposed system is better than the existing system.

22.  The results are not enough explained that help to analyze architecture can better meet the complex requirements of cross-border logistics IoT systems, reduce the risk of a single point of failure in the system, and improve system availability and reliability.

Authors needs to improve the english language of the paper. 

Author Response

Response to Reviewer 1 Comments

Comment 1: The abstract is not clearly explaining the purpose of this research.

Response 1: Thank you very much for your feedback. Your valuable comment is instrumental in improving the quality of our article. We have reorganized the language and revised the abstract to make it more concise. The details are as follows:

In cross-border transactions, the transmission and processing of logistics information directly affect the trading experience and efficiency. The use of Internet of Things (IoT) technology can make this process more intelligent, efficient, and secure. However, most traditional IoT logistics systems are provided by a single logistics company. These independent systems need to withstand high computing loads and network bandwidth when processing large-scale data. Additionally, due to the complex network environment of cross-border transactions, the platform's information security and system security are difficult to guarantee. To address these challenges, this paper designs and implements an intelligent cross-border logistics system platform that combines serverless architecture and microservice technology. This system can uniformly distribute the services of all logistics companies and divide microservices based on actual business needs. It also studies and designs corresponding Application Programming Interface (API) gateways to solve the interface exposure problem of microservices, thereby ensuring the system's security. Furthermore, asymmetric encryption technology is used in the serverless architecture to ensure the security of cross-border logistics data. The experiments show that this research solution validates the advantages of combining serverless architecture and microservices, which can significantly reduce the operating costs and system complexity of the platform in cross-border logistics scenarios. It allows for resource expansion and billing based on application program requirements at runtime. The platform can effectively improve the security of cross-border logistics service processes and meet cross-border transaction needs in terms of data security, throughput, and latency.

Comment 2: The significance of the research is not mentioned properly, instead, there is only a focus on the solution.

Response 2: Thank you for your suggestion. We understand your point of view and we have made changes to highlight the research motivation and significance of this paper in the 1.1 Motivation section. We will also make a greater effort to emphasize the research significance in our future work to enhance the quality and value of the article. Here are the specific modifications made:

The distributed nature of IoT scenarios aligns well with distributed application architectures. Microservice architecture and serverless architecture are popular software architecture design paradigms widely used in various fields as important components of distributed application architecture. Microservice architecture can be used to build more flexible and scalable systems, offering better scalability, maintainability, and extensibility compared to complex centralized processing methods, thereby better responding to changing business needs, and improving system reliability and performance. Additionally, serverless architecture provides a more flexible and agile development approach that can enable faster deployment of new applications while reducing development and operational costs.

However, microservice architecture and serverless architecture also face some challenges in terms of system integrity and information security, such as information privacy leaks, interface exposure, and low security and privacy of various microservices and functions. This paper aims to investigate the design scheme of a cross-border logistics compliance platform based on microservice and serverless architecture. Specifically, this paper will explore how to use a combination of microservice and serverless architecture to enhance the security of cross-border logistics compliance transaction platforms, while achieving the platform's design, construction, and testing. The research in this paper will help to improve the stability and security of IoT applications and provide a reference for relevant research.

Comment 3: Similar sentences are used multiple times in the research paper.

Response 3: We sincerely appreciate the reviewer's careful reading and apologize for our oversight. As suggested by the reviewer, we have reviewed the entire paper and removed similar sentences. For example:

1 Introduction:

To cope with these challenges, the cross-border logistics industry has begun to explore and apply IoT security technologies, including identity authentication, encryption technology, intrusion detection and prevention, security training, and awareness improvement. At the same time, the cross-border logistics industry has also started to pay attention to and adopt microservice and serverless architectures, by splitting the system into multiple small, independent services to achieve more efficient and secure interaction and collaboration.

  • Motivation

In the field of cross-border logistics, the IoT technology is widely used for cargo tracking, transport route optimization, and data analysis, while the security issues in the distributed IoT systems have also received increasing attention。

The advantages of microservice and serverless architectures have been demonstrated in the distributed scenario. Through the application of microservice and serverless architectures, complex business in centralized systems can be decomposed into small, independent modules。

Therefore, how to design the microservice division method and communication method between microservices reasonably, and how to effectively transmit information security in the microservice and serverless architectures, are important research contents.

Comment 4: The significance of the proposed system and how it is making a better impact than existing systems are not defined clearly.

Response 4: Thank you very much for your suggestions. As mentioned in our abstract, most traditional IoT logistics systems are provided by a single logistics company. These independent systems need to withstand high computing loads and network bandwidth when processing large-scale data. Additionally, due to the complex network environment of cross-border transactions, the platform's information security and system security are difficult to guarantee. To address these challenges, this paper designs and implements an intelligent cross-border logistics system platform that combines serverless architecture and microservice technology. This system can uniformly distribute the services of all logistics companies and divide microservices based on actual business needs. It also studies and designs corresponding Application Programming Interface (API) gateways to solve the interface exposure problem of microservices, thereby ensuring the system's security. Furthermore, asymmetric encryption technology is used in the serverless architecture to ensure the security of cross-border logistics data.

In addition, in the final paragraph of the 1.1 Motivation section, we explained the significant problem addressed by our proposed platform, as follows:

Microservice architecture and serverless architecture also face some challenges in terms of system integrity and information security, such as information privacy leaks, interface exposure, and low security and privacy of various microservices and functions. This paper aims to investigate the design scheme of a cross-border logistics compliance platform based on microservice and serverless architecture. Specifically, this paper will explore how to use a combination of microservice and serverless architecture to enhance the security of cross-border logistics compliance transaction platforms, while achieving the platform's design, construction, and testing. The research in this paper will help to improve the stability and security of IoT applications and provide a reference for relevant research.

Comment 5: The contribution section just explains the benefits of some components instead of explaining real contributions.

Response 5: Thank you very much for your suggestions. We have revised and summarized the main contributions of our study as follows:

  • The paper uses domain-driven design (DDD) method to design and divide the key microservices. The system is divided into five isolated business domains, which can isolate the impact of an attack to a specific service scope, thus avoiding single-point failure and improving the security of the entire system.
  • An API gateway is designed to address the risk of information leakage due to microservices' exposed interfaces. The API gateway provides unified interface requests and authorization filtering, ensuring communication security between microservices and clients. To protect data confidentiality during transmission, the system employs an asymmetric encryption method to encrypt sensitive data fields.
  • Experimental results demonstrate that the platform can effectively enhance system security, and meet the key performance requirements of cross-border logistics, including data security, throughput, and latency.

Comment 6: There is a need to explain the existing cross-border logistics industry working process. Also, how the proposed architecture will help in meeting the desired goal.

Response 6: We apologize for not clearly explaining this content in the text due to space limitations.

The process of cross-border logistics systems typically includes six steps: order management, warehouse management, transportation management, customs clearance management, logistics information management, and after-sales service. In addition to basic user functions, there should also be a super administrator responsible for system management.

Based on the above-mentioned requirements for the cross-border logistics system, we conducted a comprehensive analysis of the requirements and identified the related microservices in 3 System Design to meet the design goals in complex scenarios.

Comment 7: As mentioned, ensuring the security and privacy of communication between microservices and functions is a challenging process, then how it will enhance the security in the proposed solution?

Response 7: As mentioned in section 4.3 "Microservice Communication Model," we designed and constructed the communication method of the system architecture. An API gateway was designed to process distributed requests, and services are requested through asynchronous communication methods. Compared with the traditional centralized system architecture, which directly accesses the service interface, the designed system uses an API gateway to manage service requests uniformly, and the API gateway can filter interface service requests uniformly to avoid unauthorized access and message theft, thus improving the security of the system platform. And we describe more details about the API gateway in our manuscript:

To address the aforementioned issues, the system design employs an API gateway to encapsulate the microservice APIs[52]. The gateway intercepts all incoming request data and forwards the requests to backend services. The architecture of the API gateway is illustrated in Figure 10, and its functionality is described as follows:

  1. Request routing: all client requests first arrive at the API gateway, which queries the route mapping and forwards the request to the corresponding backend service. This function serves as a reverse proxy for backend microservices.
  2. Protocol conversion: client requests are often in the form of HTTP-based RESTful requests, while backend services may use gRPC. In this case, the API gateway can perform protocol conversion, reducing client implementation costs.
  3. Authentication and authorization: the API gateway determines client access permissions by verifying client request identity.
  4. Speed limit: limits the number of client requests per second, reducing system pressure.
  5. Log monitoring: important API requests can be logged and monitored to further enhance system security.

Through the use of the API gateway, clients only need to send requests to the gateway, without worrying about routing, protocol, and other issues, reducing the difficulty of client implementation. Furthermore, backend services can easily implement functions such as authentication, authorization, speed limiting, and monitoring, reducing system coupling and improving system security.

Comment 8: All the terms in related work, are defined explicitly and there is no comparison of existing systems with the proposed system.

Response 8: Thank you very much for your suggestion. In the Related Work section, we mainly provided a brief summary of the current IoT security solutions, IoT resource allocation schemes, and existing architectures of distributed systems. To clearly describe this section, we have included a summary table. The proposed system in this paper is closely related to these three research areas. However, since there are few platform design schemes that consider all three aspects at the same time, we regret that we have not found a suitable platform to compare with our proposed platform design.

Purpose

References

Methods

Advantages

Disadvantages

Security of IoT

[8], [9], [10], [11], [13], [14], [15], [16], [19], [20]

Blockchain,

Security Protocol,

Edge Computing.

Improve the Security of the Internet of Things Communication

High Computation and Communication Overhead

Resource Allocation of IoT

[25], [26], [27], [29], [31], [32], [37], [38]

Dynamic Choreography of Edge Computing, Cloud Computing, Fog Computing

High Real-time Performance, Low Network Pressure

High Deployment Overhead, Poor Scalability

Distributed Architecture

[41], [42], [46]

Service-Oriented Architecture Microservice architecture, Serverless Technology

High Development Efficiency, Lightweight and Low Cost

Maintenance Difficulty, Business Splitting and Decentralization

Table 1. Summary of Related Work.

Comment 9: The system design is explained generically, however, it is lacking in explaining the application of system design to the proposed work.

Response 9: Thank you for your suggestion. Unlike traditional monolithic architecture, microservices architecture requires corresponding microservices division based on the system application design, before proceeding with the system implementation. Therefore, in System Design section, we mainly focused on key microservices division for cross-border logistics scenarios:

According to the results of the bounded context, the microservices can be split. In theory, one bounded context corresponds to one microservice, but factors such as service responsibility and team heterogeneity need to be considered during the application process. In this system, the results of the bounded context are directly used as the basis for microservice division. The final division of microservices includes three core domains: user domain, order domain, and rule domain. In addition to these three core domains, there is also a statistics domain used for satisfying administrator information retrieval, and a general domain used for system support. The detailed introduction of all microservices domains in the system can be found in Table 2.

Microservice Name

Function Description

User Domain

Responsible for user management, including registration, login, add, forget password, etc.

Rule Domain

Manage contents related to product compliance requirements

Order Domain

Requests sent by sellers to service providers as order types

Monitor Domain

Collection and statistics of user request access data, providing functional view for administers.

Common Domain

Services open to all domains, including mail delivery, message queue and other facilities.

In the fourth section of System Implementation, we provided detailed exposition and explanation of the proposed system solution:

Based on the design and partitioning of microservices, we have detailed the overall architecture of the platform. In this section, we present the system platform that combines microservice and Serverless architecture through various aspects such as the overall architecture diagram, system aspect diagram, process view, microservice communication mode, and microservice interaction design. The system architecture design introduces in detail the overall technical architecture of the system, the layered service logic structure, and the data flow in the system. The continuous integration/continuous deployment design details the deployment methods of the frontend and backend and the detailed steps involved. The microservice communication mode elaborates on the platform's technology selection reasons and the proposed API gateway solution.

Comment 10: The whole paper is explaining new terms, instead of aligning the concept with the problem. 

Response 10: Thank you for your careful review of our research. We apologize that the organization of our paper may have caused confusion for you. Our study is divided into six parts. The first part provides a detailed explanation of the research motivation and challenges. The second part describes and summarizes the related work involved. The third part uses domain-driven design to identify and divide the microservices involved in cross-border logistics. Based on the microservice division in the third part, we design and describe the key aspects of the platform's overall front-end and back-end architecture, microservice communication method, system storage method, etc. in the fourth part. We then use this as a basis to build the platform. In the fifth part, we use testing tools to systematically test and discuss the platform we have built. The final part provides a summary of the entire study.

Comment 11: The paper needs attention on compiling the concepts in a better way, so, the reader remains concentrated on the sections that are leading towards the solution.

Response 11: Thank you for your suggestion. We have focused our introduction to conceptual content in the Introduction and Related Work sections, and we have revised and updated this section accordingly. The third section and subsequent content are detailed descriptions of the system platform design proposed in this study for the cross-border logistics scenario.

Comment 12: There is a need to rephrase some statements so they depict a clear image of the proposed system.

Response 12: Thank you for your suggestion, it was very helpful for us to revise the paper. We have added a description of the overall system in section 4.1 System Architecture Design, and Figure 4 also shows the overall structure of the proposed system platform. Some details are as follows:

In this section, we present the system platform that combines microservice and Serverless architecture through various aspects such as the overall architecture diagram, system aspect diagram, process view, microservice communication mode, and microservice interaction design. The system architecture design introduces in detail the overall technical architecture of the system, the layered service logic structure, and the data flow in the system. The continuous integration/continuous deployment design details the deployment methods of the frontend and backend and the detailed steps involved. The microservice communication mode elaborates on the platform's technology selection reasons and the proposed API gateway solution.

Comment 13: There is a need to discuss the technological advancements you have added to your research.

Response 13: Thank you for your suggestion. The front-end and back-end development of the system designed in this article adopt the CI/CD pipeline technology, which automates the integration, building, testing, and deployment of the code, thereby reducing the time for manual operations and greatly improving the efficiency and quality of front-end and back-end development. It also reduces the chances of errors and the cost of fixing them, while also reducing the complexity and risk of the system.

Comment 14: Multiple terms are mentioned in the paper with an abbreviation, so try to mention the complete phrase.

Response 14: Thank you for your suggestion. We apologize for our oversight. In the latest version, we have reviewed the entire document and made modifications to the abbreviations used in the paper.

Comment 15: Mention at least a one-paragraph description of software or tools that are used in the research paper. As in system architecture, there are so many tools used, however, in the whole paper, not a single statement explains the purpose of using them.

Response 15: Thank you for your suggestion. We have updated the description of the tools used in section 4.1 System Architecture Design as follows:

The system's technical architecture mainly consists of the following components:

  • Microservice backend module composed of Lambdas

The backend adopts AWS Lambda as the computing unit to implement a serverless system architecture with automatic scaling and pay-per-use advantages.

  • Frontend module built using React and hosted on cloud services

React is used to develop the frontend functional modules, and the built files are deployed to S3 and distributed through CloudFront CDN to accelerate global user access.

  • Serverless general component services provided by cloud service providers

The general serverless components include DynamoDB NoSQL databases, message queues, email services, and object storage services. Except for the database, which is dedicated to the microservice, all other components exist in the form of general components that can be used by the microservice computing module (Lambda).

  • Continuous integration and continuous deployment (CI/CD) pipelines

The CI/CD pipeline is divided into two independent pipelines for the frontend and backend. The pipeline includes the code repository, build service, and deployment unit. When changes are detected in the main branch of the code repository, the pipeline starts to operate, and after approval, it deploys to the account environment. Adopting CI/CD can accelerate development speed and reduce delivery time.

Additionally, we have added an "Experimental Environment" section in the "5 Results and Discussion" part to introduce the testing experimental environment and tools for the system. The specific content is as follows:

5.1 Experimental Environment

Taurus is a tool for running various open-source load and functional tests. In the experiment, two Ubuntu20.04 system hosts were set up, and Taurus was used on one device to test the platform designed in this study installed on the other.

Comment 16: The result and Discussion section is so brief, as it not explaining the outcomes and evaluation parameters.

Response 16: Thank you for your suggestion. We have revised the section of "5. Results and Discussions" and provided a detailed explanation of the system testing conducted and the corresponding discussions. The specific modifications are as follows:

The testing in this study was conducted using two methods: load testing and stress testing.

The load testing method involved using Taurus to interact with the system's API interface. By simulating user requests, the API return results were obtained in what is known as a single request. The server was subjected to stress load testing by increasing the number of servers and Taurus threads. Figure 15 shows the distributed test output using Taurus with 100 servers and 100 threads per server. In the scenario of a large number of distributed request loads within a short period, the self-scaling system proposed in this study expanded the bandwidth resources to 1.98 Kps, with an average latency of 1.82070s and a system response error rate of 0.0098%. The comprehensive test results indicate that the system we designed is capable of maintaining high performance, reliability, stability, and information processing capability when facing a large number of requests.

During stress testing, a large number of concurrent users or requests are simulated using tools to evaluate the system's ability to handle a heavy load and maintain key performance metrics, such as response time, availability, and scalability during peak loads. As shown in Table 3, the testing results reveal the system's response time and success rate metrics when the number of testing servers ranges from 1 to 50. When 100 servers are deployed, each with 100 threads, the system's peak RPS is achieved, processing 5425 requests per second. However, the average response time of requests is around 1.8 seconds, attributed to network congestion caused by the excessive number of Taurus threads simultaneously sending requests from a single testing server.

Through CloudWatch monitoring, the growth curves of the backend Lambda invocation and instance scaling are shown in Figure 16. As the number of requests increases, Lambda automatically scales its service instances to handle the large volume of requests. The system proposed in this paper performs self-scaling of service resources based on actual request traffic and environmental demands, avoiding resource shortages or waste caused by rigid, centralized software architectures.

Comment 17: The result and discussion section is not summarizing the proposed solution efficiently.

Response 17: Thank you for your suggestion. In the Results and Discussion section, we focused on the experimental testing and discussion of the proposed system platform. We also added a paragraph at the end of this section to summarize the proposed solution. The specific content added is as follows:

In this section, the advantage of combining Serverless architecture with microservice framework in automatic scaling is verified through load testing and stress testing. The test results demonstrate that the system is able to expand computing resources according to the number of requests without any operational intervention, and the throughput can meet the needs of most businesses. Moreover, in the event of attacks such as denial of service due to a large number of requests, the platform we designed can also avoid single point of failure and meet the requirements of system stability and security.

Comment 18: The result section is not explaining the evaluation method used to find out whether the proposed system is meeting the development needs.

Response 18: Thank you for your constructive feedback, it has greatly helped us improve the quality of our article. We have added a detailed explanation and summary of the testing method used to verify that the proposed system meets the development needs in the final paragraph of the Result and Discussion section.

Comment 19: There is a single table in the result section that is describing the load testing. There are other parameters such as data security and latency, and how it is improved is missing in the discussion.

Response 19: Thank you very much for your suggestion. Table 3 is only used for analyzing the results of the load test. Regarding the latency parameter you mentioned, you can see the test results in Figure 13, from which we can obtain detailed data such as the average latency, average bandwidth, and average response time during the system test. In addition, we are sorry that due to the limitations of the current testing environment, we have not yet completed security testing of the system. We are expected to make efforts to set up a security testing environment and conduct security testing of the system platform in the future.

Comment 20: No real-life scenario or evaluation measures are used to find out the efficiency of the proposed solution.

Response 20: Thank you for your valuable comment. In the Result and Discussion section, we first built the software platform based on the design proposal of the implementation part. On this basis, we conducted system testing using common testing tools. As stated in this section, our proposed system platform design can meet the various functional requirements of cross-border logistics transportation. At the same time, the system can expand computing resources according to the number of requests without any operation and maintenance intervention, and its throughput can meet most business needs. In the face of a large number of request denial of service attacks, our designed platform can also avoid single point of failure and meet the requirements of system stability and security.

Comment 21: There is a need to explain how the proposed system is better than the existing system.

Response 21: Thank you for your valuable comments. Our proposed system is mainly aimed at the compliant system platform in the context of cross-border logistics. Currently, most of the existing logistics platforms are developed by logistics companies themselves, and their systems are constructed and managed by the companies themselves. At the same time, their business is also limited to designated countries and designated policies. However, the construction and management of independent company platforms are difficult to adapt to the complex business flow and constantly changing national policies under economic globalization. Therefore, this paper completed the design of a cross-border logistics compliant system based on microservices and serverless architecture, greatly facilitating logistics users to find the compliance policies of target products in the destination market and finding corresponding compliant evaluation merchants and logistics companies to request inspection and transportation. This solves the problems faced by traditional logistics platforms such as a single evaluation policy and single logistics merchant. At the same time, this paper explores a new mode of architectural design combining microservices and serverless architecture. The designed platform has security and stability, improves the operating safety of the system, and reduces the resource cost of the platform. At the same time, in large-scale cross-border logistics scenarios, the system has sufficient throughput capacity and can meet real-time responses to a large number of requests.

Comment 22: The results are not enough explained that help to analyze architecture can better meet the complex requirements of cross-border logistics IoT systems, reduce the risk of a single point of failure in the system, and improve system availability and reliability.

Response 22: Thank you for raising this question. When conducting the test analysis, we mainly focused on the automatic scaling ability of the Serverless architecture and the load capacity of the application. To address the complex requirements of your cross-border logistics IoT system, the risk of single-point failure, as well as the improvement of availability and reliability, we have used asynchronous message queues in the 4.3 microservice communication mode design to ensure task sequencing and processing integrity, in order to improve the reliability and durability of the system. Regarding security, we have implemented permission and request filtering through an API gateway to enhance the system's security protection capability.

Reviewer 2 Report

1. The Abstract should be very precise.

2. The Introduction section is very poor. In a research article, the introduction section must be very strong with the motivations of this paper, which is missing in this paper. Moreover, the disadvantages of the existing schemes must be discussed to motivate this new work.

3. The mentioned point-wise contributions should be very precise. Dont give long lines. 

4. The Related Work section is poor. The authors are suggested not to give many subsections. Some related schemes can be discussed in this section. The authors must include some more recent schemes. Also, the following papers must be cited to improve this section, as well as the Reference section:

a) SDN-based intrusion detection system for IoT using deep learning classifier (IDSIoT-SDL)

b) Revisiting shift cipher technique for amplified data security

c) Introduction to the special section on advances of machine learning in cybersecurity (VSI-mlsec)

d) Boosting image watermarking authenticity spreading secrecy from counting-based secret-sharing

e) Blockchain-based medical certificate generation and verification for IoT-based healthcare systems

f) Research on internet security situation awareness prediction technology based on improved RBF neural network algorithm

g) A new table based protocol for data accessing in cloud computing

5. In Section 2, a table can be given to summarize the entire section.

6. Why Domain-Driven Design is popular?

7. Threat model is not clear. Which attacks are considered and why?

8. Figure 1 is completely meaningless.

9. Search stage is completely unclear.

10. Which entities are involved for key management?

11. Discuss front-end CI/CD streamline design process properly. 

12. In section 5, add the “Experimental Environment” section.

13. How Taurus Test has been performed? Mention clearly.

14. How the results of Figure 14 and Table 3 are generated? The caption style of the figure is wrong.

15. Technical details about results are missing.

16. Which existing schemes are used for performance analysis and why?

17. What is the novelty of this work? It is hard to identify from the current version of this paper.

18. Many figures are given in this paper. But, there are no linkage in the concept.

19. The organization of the paper must be improved. The paper must be formatted properly.

20. Improve the English language.

21. The Reference section must be improved significantly.

22. Use a well-known software to draw the diagrams. 

The English language must be improved.

Author Response

Response to Reviewer 2 Comments

Comment 1: The Abstract should be very precise.

Response 1: Thank you very much for your feedback and suggestions. Based on your comments, we have revised the abstract to make it more concise and clear, highlighting the key points of the paper. The updated abstract is as follows:

We have rephrased and condensed the language in the abstract. The revised version is as follows:

In cross-border transactions, the transmission and processing of logistics information directly affect the trading experience and efficiency. The use of Internet of Things (IoT) technology can make this process more intelligent, efficient, and secure. However, most traditional IoT logistics systems are provided by a single logistics company. These independent systems need to withstand high computing loads and network bandwidth when processing large-scale data. Additionally, due to the complex network environment of cross-border transactions, the platform's information security and system security are difficult to guarantee. To address these challenges, this paper designs and implements an intelligent cross-border logistics system platform that combines serverless architecture and microservice technology. This system can uniformly distribute the services of all logistics companies and divide microservices based on actual business needs. It also studies and designs corresponding Application Programming Interface (API) gateways to solve the interface exposure problem of microservices, thereby ensuring the system's security. Furthermore, asymmetric encryption technology is used in the serverless architecture to ensure the security of cross-border logistics data. The experiments show that this research solution validates the advantages of combining serverless architecture and microservices, which can significantly reduce the operating costs and system complexity of the platform in cross-border logistics scenarios. It allows for resource expansion and billing based on application program requirements at runtime. The platform can effectively improve the security of cross-border logistics service processes and meet cross-border transaction needs in terms of data security, throughput, and latency.

Comment 2: The Introduction section is very poor. In a research article, the introduction section must be very strong with the motivations of this paper, which is missing in this paper. Moreover, the disadvantages of the existing schemes must be discussed to motivate this new work.

Response 2: Thank you very much for your feedback. We understand your opinion and agree on the importance of the introduction section in a research article. We have made modifications to the 1.1 Motivation section to highlight the research motivation and the shortcomings of existing technological solutions, and we will make a greater effort to emphasize the importance of the introduction section in our future work to enhance the quality and value of our article. Thank you again for your valuable suggestions. The specific modifications are as follows:

The distributed nature of IoT scenarios aligns well with distributed application architectures. Microservice architecture and serverless architecture are popular software architecture design paradigms widely used in various fields as important components of distributed application architecture. Microservice architecture can be used to build more flexible and scalable systems, offering better scalability, maintainability, and extensibility compared to complex centralized processing methods, thereby better responding to changing business needs, and improving system reliability and performance. Additionally, serverless architecture provides a more flexible and agile development approach that can enable faster deployment of new applications while reducing development and operational costs.

However, microservice architecture and serverless architecture also face some challenges in terms of system integrity and information security, such as information privacy leaks, interface exposure, and low security and privacy of various microservices and functions. This paper aims to investigate the design scheme of a cross-border logistics compliance platform based on microservice and serverless architecture. Specifically, this paper will explore how to use a combination of microservice and serverless architecture to enhance the security of cross-border logistics compliance transaction platforms, while achieving the platform's design, construction, and testing. The research in this paper will help to improve the stability and security of IoT applications and provide a reference for relevant research.

Comment 3: The mentioned point-wise contributions should be very precise. Dont give long lines. 

Response 3: Thank you for your suggestion. We have revised and summarized the main contributions of our research. The specific content is as follows:

  • The paper uses domain-driven design (DDD) method to design and divide the key microservices. The system is divided into five isolated business domains, which can isolate the impact of an attack to a specific service scope, thus avoiding single-point failure and improving the security of the entire system.
  • An API gateway is designed to address the risk of information leakage due to microservices' exposed interfaces. The API gateway provides unified interface requests and authorization filtering, ensuring communication security between microservices and clients. To protect data confidentiality during transmission, the system employs an asymmetric encryption method to encrypt sensitive data fields.
  • Experimental results demonstrate that the platform can effectively enhance system security, and meet the key performance requirements of cross-border logistics, including data security, throughput, and latency.

Comment 4: The Related Work section is poor. The authors are suggested not to give many subsections. Some related schemes can be discussed in this section. The authors must include some more recent schemes. Also, the following papers must be cited to improve this section, as well as the Reference section:

  1. a) SDN-based intrusion detection system for IoT using deep learning classifier (IDSIoT-SDL)
  2. b) Revisiting shift cipher technique for amplified data security
  3. c) Introduction to the special section on advances of machine learning in cybersecurity (VSI-mlsec)
  4. d) Boosting image watermarking authenticity spreading secrecy from counting-based secret-sharing
  5. e) Blockchain-based medical certificate generation and verification for IoT-based healthcare systems
  6. f) Research on internet security situation awareness prediction technology based on improved RBF neural network algorithm
  7. g) A new table based protocol for data accessing in cloud computing

Response 4: Thank you for your suggestion. We have revised the Related Work section and added some references, such as: [5], [6], [7], [17], [45].

  • SDN-based intrusion detection system for IoT using deep learning classifier (IDSIoT-SDL).
  • Blockchain-Based Medical Certificate Generation and Verification for IoT-Based Healthcare Systems.
  • Research on Internet Security Situation Awareness Prediction Technology Based on Improved RBF Neural Network Algorithm.
  • Introduction to the Special Section on Advances of Machine Learning in Cybersecurity (VSI-Mlsec).

[45] A New Table Based Protocol for Data Accessing in Cloud Computing.

Since the other two references are not very relevant to the content of this article, we have not added them to the article.

Comment 5: In Section 2, a table can be given to summarize the entire section.

Response 5 : Thank you for your constructive feedback, it has greatly helped us improve the quality of our article. We added a table to summarize the entire section.

Purpose

References

Methods

Advantages

Disadvantages

Security of IoT

[8], [9], [10], [11], [13], [14], [15], [16], [19], [20]

Blockchain,

Security Protocol,

Edge Computing.

Improve the Security of the Internet of Things Communication

High Computation and Communication Overhead

Resource Allocation of IoT

[25], [26], [27], [29], [31], [32], [37], [38]

Dynamic Choreography of Edge Computing, Cloud Computing, Fog Computing

High Real-time Performance, Low Network Pressure

High Deployment Overhead, Poor Scalability

Distributed Architecture

[41], [42], [46]

Service-Oriented Architecture Microservice architecture, Serverless Technology

High Development Efficiency, Lightweight and Low Cost

Maintenance Difficulty, Business Splitting and Decentralization

Comment 6: Why Domain-Driven Design is popular?

Response 6: Domain-Driven Design has become popular because it provides an effective approach to building high-quality software systems that combine software design with business requirements in complex business scenarios. This design emphasizes understanding and analyzing the business domain, and by thoroughly understanding business requirements and processes, better systems can be designed that meet business needs. Additionally, this design places emphasis on the design of the business domain and domain model, which can better control the complexity and design quality of the system, thereby improving the system's maintainability, scalability, and testability.

Comment 7: Threat model is not clear. Which attacks are considered and why?

Response 7: Thank you for your feedback. We mainly consider malicious node attacks, information disclosure attacks, and remote control attacks. Attackers can manipulate or forge sensor data by controlling or disguising themselves as legitimate IoT nodes, and steal sensitive information transmitted in the IoT network, such as user privacy data, confidential business information, etc., using various means such as network sniffing and port scanning. Or by attacking the remote control interface of IoT devices, they can control the devices and execute malicious commands. The solution proposed in this paper, based on Serverless and microservices architecture, can isolate attacks within the affected service scope by separating and isolating different service systems, thus ensuring the security of the entire system.

Comment 8: Figure 1 is completely meaningless.

Response 8: Thank you for your suggestion. To design a system solution that meets the requirements of cross-border transportation services, it is necessary to use domain-driven design (DDD) to make specific and reasonable divisions of microservices based on the application domain characteristics of the system platform. Therefore, the domain design results shown in Figure 1 are essential.

Comment 9: Search stage is completely unclear.

Response 9: We apologize for any confusion caused by the organization of our paper. This study is divided into six parts. The first part provides a detailed explanation of the research motivation and challenges. The second part describes and summarizes the relevant work involved. In the third part, we use the domain-driven design method to analyze and divide the microservices involved in cross-border logistics. Based on the microservice division design in the third part, we design and describe key aspects of the platform's overall front-end and back-end architecture, microservice communication methods, and system storage methods in the fourth part. We then use this as a basis to build the platform. In the fifth part, we use testing tools to perform system testing and discussion on the platform we have built. Finally, the last part summarizes the entire research paper.

Comment 10: Which entities are involved for key management?

Response 10: We apologize for not elaborating on asymmetric encryption in the text and only briefly mentioning it. In the system design proposed in this paper, we use the AWS KMS asymmetric encryption method to store the keys and encrypt sensitive data fields to ensure data security. The process of encrypting data using KMS is illustrated in the following figure. The process is described as follows:

Figure 12. KMS encrypt

  • Request a new data key under the customer master key (CMK). Receive the encrypted data key and the plaintext version of the data key.
  • In the AWS Encryption SDK, the plaintext data key is used to encrypt the message. The plaintext data key is then deleted from memory.
  • The encrypted data key and the encrypted message are combined into a single ciphertext byte array.

The process of decrypting data using KMS is illustrated in Figure 13 and described as follows:

Figure 13.  KMS decrypt

(1) Analyze the encrypted message in the envelope to obtain the encrypted data key, and request the AWS Encryption SDK to decrypt the data key.

(2) Receive the plaintext data key from the AWS Encryption SDK.

(3) Use the data key to decrypt the message and return the original plaintext.

Comment 11: Discuss front-end CI/CD streamline design process properly. 

Response 11: We apologize for the unclear wording in our paper. The front-end CI/CD section was mainly intended to provide a brief overview of the process for the platform we designed, and did not require a detailed explanation of the CI/CD process itself that would take up a significant amount of space.

Comment 12: In section 5, add the “Experimental Environment” section.

Response 12: Thank you very much for your suggestion. Your advice is very helpful in improving the quality of our paper. We have added an "Experimental Environment" section to the latest manuscript. The specific content is as follows:

5.1 Experimental Environment

Taurus is a tool for running various open-source load and functional tests. The experiment set up two Ubuntu 20.04 systems. One of the devices used the Taurus testing tool to test the platform designed in this study. For the load testing tool Taurus, a pre-packaged distributed load testing solution is available, which encapsulates Taurus as a container and can be easily deployed to multiple test servers. Test parameters such as the number of test servers, threads, target APIs, duration, etc. can be set through the front-end page. The entire testing process is shown in Figure 14.

Figure 14. Taurus Distributed Load Test Solution.

Comment 13: How Taurus Test has been performed? Mention clearly.

Response 13: Thank you for your valuable suggestion. We have added a paragraph in the section "5. Results and Discussions" to explain how the testing was conducted using Taurus. The specific content is as follows:The testing method involved Taurus making requests to the system's API interface. By simulating user requests, Taurus obtained API response results, which is referred to as a single request. The server was subjected to stress load testing by increasing the number of servers and Taurus threads.

Comment 14: How the results of Figure 14 and Table 3 are generated? The caption style of the figure is wrong.

Response 14: We apologize for the poor presentation of our images. As we described in the third paragraph of section 5, Results and Discussions, we conducted system testing using the Taurus tool. Figures 14(new version Figure 16) and Table 3(new version Table 4) are real experimental data charts obtained through the testing tool.

Comment 15: Technical details about results are missing.

Response 15: Thank you very much for your suggestion. As you pointed out, our description of the test results was too vague. Therefore, in section 5.2, we provided a detailed description of the testing methods and system test results. The specific content is as follows:

The testing in this study was conducted using two methods: load testing and stress testing.

The load testing method involved using Taurus to interact with the system's API interface. By simulating user requests, the API return results were obtained in what is known as a single request. The server was subjected to stress load testing by increasing the number of servers and Taurus threads. Figure 15 shows the distributed test output using Taurus with 100 servers and 100 threads per server. In the scenario of a large number of distributed request loads within a short period, the self-scaling system proposed in this study expanded the bandwidth resources to 1.98 Kps, with an average latency of 1.82070s and a system response error rate of 0.0098%. The comprehensive test results indicate that the system we designed is capable of maintaining high performance, reliability, stability, and information processing capability when facing a large number of requests.

Figure 15. Taurus Test Outputs

During stress testing, a large number of concurrent users or requests are simulated using tools to evaluate the system's ability to handle a heavy load and maintain key performance metrics, such as response time, availability, and scalability during peak loads. As shown in Table 3, the testing results reveal the system's response time and success rate metrics when the number of testing servers ranges from 1 to 50. When 100 servers are deployed, each with 100 threads, the system's peak RPS is achieved, processing 5425 requests per second. However, the average response time of requests is around 1.8 seconds, attributed to network congestion caused by the excessive number of Taurus threads simultaneously sending requests from a single testing server.

Through CloudWatch monitoring, the growth curves of the backend Lambda invocation and instance scaling are shown in Figure 16. As the number of requests increases, Lambda automatically scales its service instances to handle the large volume of requests. The system proposed in this paper performs self-scaling of service resources based on actual request traffic and environmental demands, avoiding resource shortages or waste caused by rigid, centralized software architectures.

Table 4. Load Test.

Number of Servers

Number of Concurrent Threads

Throughput

Average Response Time

Total Number of Requests

Number of Failed Requests

1

100

49.8/s

1.97874 s

29830

0

5

100

268.52/s

1.8458 s

160843

0

10

100

549.16/s

1.79907 s

329493

0

50

100

2698.09/s

1.82738 s

1618854

0

100

100

5425.95/s

1.82138 s

3250144

3

Figure 16. Invocations and Concurrent.

In this section, the advantages of combining Serverless architecture and microservice framework with automatic scaling were verified through load testing and stress testing. The test results showed that the system was able to expand computing resources based on the number of requests without any operational intervention, and the throughput was able to meet the needs of most businesses. When faced with attacks such as denial of service due to a large number of requests, the platform we designed is also able to avoid single points of failure, meeting the requirements for stability and security of the system.

Comment 16: Which existing schemes are used for performance analysis and why?

Response 16: In this system, due to the involvement of a large number of devices, sensors, network communication, and data processing, extensive system testing is required to verify its performance, stability, and reliability. Taurus is an open-source automated testing tool that can be used to load test and performance test various types of applications, which is very suitable for this system. Therefore, we used this testing tool to conduct system testing on the platform system designed in this paper.

Comment 17: What is the novelty of this work? It is hard to identify from the current version of this paper.

Response 17: We apologize for not clearly describing the novelty of our research in the paper. In the contribution section, we summarized the innovation of our research in the design of a secure platform for cross-border logistics IoT systems. The specific content is as follows:

  • The paper uses domain-driven design (DDD) method to design and divide the key microservices. The system is divided into five isolated business domains, which can isolate the impact of an attack to a specific service scope, thus avoiding single-point failure and improving the security of the entire system.
  • An API gateway is designed to address the risk of information leakage due to microservices' exposed interfaces. The API gateway provides unified interface requests and authorization filtering, ensuring communication security between microservices and clients. To protect data confidentiality during transmission, the system employs an asymmetric encryption method to encrypt sensitive data fields.
  • Experimental results demonstrate that the platform can effectively enhance system security, and meet the key performance requirements of cross-border logistics, including data security, throughput, and latency.

Comment 18: Many figures are given in this paper. But, there are no linkage in the concept.

Response 18: The proposed solution based on Serverless and microservices architecture in this paper aims to improve the security and scalability of IoT applications by separating and isolating different services. The provided images are designed for the system platform functions and architecture of a specific independent module or function of the IoT security system in the context of cross-border logistics, so the architecture diagrams of each module are not closely related to each other.

Comment 19: The organization of the paper must be improved. The paper must be formatted properly.

Response 19: Thank you very much for your valuable suggestions. We have reviewed and updated our content throughout the paper.

Comment 20: Improve the English language.

Response 20: Thank you for your suggestion. We have made every effort to polish the language in the revised manuscript.And we hope the revised manuscript could be acceptable for you.

Comment 21: The Reference section must be improved significantly.

Response 21: We sincerely appreciate the valuable comments. We have checked the literature carefully and added references into the Related Work part in the revised manuscript.

Comment 22: Use a well-known software to draw the diagrams. 

Response 22: Thank you for your suggestion. In fact, except for Figure 15 and Figure 16, all of our images were drawn using Visio. Since Figure 15 and Figure 16 were obtained from professional testing software, we believe that using real experimental data images is more convincing.

Reviewer 3 Report

The fact that the study focuses on a serverless architecture and a microservice architecture, both of which can be utilized to build a safe Internet of Things system, is a significant point. While taking into consideration that Microservice Architecture, for Service Isolation, in which the Microservices can assist in isolating individual services within the IoT system, hence making it simpler to secure each service individually. This will be a significant contribution; however, sadly, there is a great deal of clarifications that need to be made concerning programming design associated to the research focus as follows:

·         As far as the study is concerned, the "designs" were not rooted by the fundamental research approach, which is where the technology involved needs to be supplied and analyzed.

·         There is a requirement for the presentation of computational-related cost/effect in the "Clarify User Vision" approaches of the design in section 3.1.1 from section 3.1.4 to section 3.1.6 the authors do not provide anything associated to the computing-grounded approach toward design. These techniques need to be highlighted in the form of iterative programming steps, where each computing resource that is necessary is highlighted.

·         Similarly, the design of Section 3.2 ought to also be in the form of iterative programming stages, with an emphasis on bringing attention to all of the necessary computational resources.

·         If all of the iterative programming steps had been presented in section 3, along with a highlight of each and every computational resource that was required, section 4.1 would have been far simpler. Unfortunately, because they are not offered, currently at this stage (section 4) there is a requirement for iterative programming steps, where every computational resource that is necessary for both the frontend and the backend of the Serverless architecture are highlighted.

·         The specifics of the iterative programming stages involved in the Serverless architecture, which include all of the computational resources needed for the frontend and the backend and should be described in sections 4.1.1 and 4.1.2 in details by either providing pseudocodes and give details of how they work or provide the theoretical underpinning of the approach in forms of mathematical equations.

·         Both subsection 4.2.1 and section 4.2.2 of section 4.2 are required to have specifics regarding the algorithm that is involved as well as the iterative programming stages. This is part of the section where each computing resource that is necessary for both the frontend and backend of the architecture should be emphasized.

The discussion of the results, as well as the analysis of the experiments, are not very clear. Because you supplied in Figure 12 the entire scenarios involve in your work, there is also a need to provide each experimental scenario separately along with its own outcome separately.

The explanation of how Figure 13 is interpreted is not very good, and Figure 13 itself is not very clear.

There is a significant lack of depth in the description of the outcome in Figure 14, and there is a pressing need to bring attention to the specifics of those descriptions.

The use of lexical expressions is strong; nonetheless, in order to enhance one's command of the English language, it would be beneficial to emphasize certain aspects of sentence structure.

Author Response

Response to Reviewer 3 Comments

Comment 1: The fact that the study focuses on a serverless architecture and a microservice architecture, both of which can be utilized to build a safe Internet of Things system, is a significant point. While taking into consideration that Microservice Architecture, for Service Isolation, in which the Microservices can assist in isolating individual services within the IoT system, hence making it simpler to secure each service individually. This will be a significant contribution; however, sadly, there is a great deal of clarifications that need to be made concerning programming design associated to the research focus as follows:

  • As far as the study is concerned, the "designs" were not rooted by the fundamental research approach, which is where the technology involved needs to be supplied and analyzed.
  • There is a requirement for the presentation of computational-related cost/effect in the "Clarify User Vision" approaches of the design in section 3.1.1 from section 3.1.4 to section 3.1.6 the authors do not provide anything associated to the computing-grounded approach toward design. These techniques need to be highlighted in the form of iterative programming steps, where each computing resource that is necessary is highlighted.
  • Similarly, the design of Section 3.2 ought to also be in the form of iterative programming stages, with an emphasis on bringing attention to all of the necessary computational resources.
  • If all of the iterative programming steps had been presented in section 3, along with a highlight of each and every computational resource that was required, section 4.1 would have been far simpler. Unfortunately, because they are not offered, currently at this stage (section 4) there is a requirement for iterative programming steps, where every computational resource that is necessary for both the frontend and the backend of the Serverless architecture are highlighted.
  • The specifics of the iterative programming stages involved in the Serverless architecture, which include all of the computational resources needed for the frontend and the backend and should be described in sections 4.1.1 and 4.1.2 in details by either providing pseudocodes and give details of how they work or provide the theoretical underpinning of the approach in forms of mathematical equations.
  • Both subsection 4.2.1 and section 4.2.2 of section 4.2 are required to have specifics regarding the algorithm that is involved as well as the iterative programming stages. This is part of the section where each computing resource that is necessary for both the frontend and backend of the architecture should be emphasized.

Response 1: Thank you for your feedback. We apologize for not clearly explaining this content in the text due to space limitations. The system designed in this article has many functional modules, and we focused on introducing the overall module of the system in our writing. Therefore, we did not spend too much space explaining the specific technical implementation of each module.

  • In section 3.1.1, we design pseudocode for some business requirement modules to introduce the implementation steps of each requirement module. Some details about pseudocode for the design of some requirement modules are as follows:

Requirement A: Cross-border logistics customers are able to find the comprehensive compliance requirements needed for their product target market based on finer granularity, and are able to perform self-inspection during transportation.

// Define a function to retrieve the compliance requirements of a product's target market

function getRegulations(targetMarket) {

// Search the compliance requirements database for the target market's requirements

regulations = database.search(targetMarket)

return regulations }

// Define a function for conducting self-checks

function selfCheck(shipmentInfo, regulations) {

// Check if shipment information

is compliant with the regulations isCompliant = checkCompliance(shipmentInfo, regulations)

return isCompliant }

// Implement the following steps in the system platform:

// 1. Retrieve the product's target market information

targetMarket = getInput()

// 2. Retrieve the compliance requirements of the target market

regulations = getRegulations(targetMarket)

// 3. Display the compliance requirements to the customer

displayRegulations(regulations)

// 4. Allow the customer to input shipment information and conduct self-checks

shipmentInfo = getShipmentInfo()

isCompliant = selfCheck(shipmentInfo, regulations)

// 5. Display the self-check results to the customer

displayComplianceResult(isCompliant)

Requirement B: Cross-border logistics customers can use this system platform to find suitable compliance assessment service providers and contact them for support.

// Define a function to search for compliance assessment service providers.

function searchProviders(keyword) {

// Search for service providers containing the keyword in the service provider database

providers = database.search(keyword)

return providers}

// Define a function to contact the service provider

function contactProvider(provider, customerInfo) {

// Send a request to the service provider for support

response = provider.sendRequest(customerInfo)

return response}

// Implement the following steps in the system platform

// 1. Obtain customer demand information

customerInfo = getInput()

// 2. Search for service providers that meet customer needs

providers = searchProviders(customerInfo.keywords)

// 3. Display the list of service providers for customers to choose from

displayProviders(providers)

// 4. The customer selects and contacts the appropriate service provider

selectedProvider = getSelectedProvider()

response = contactProvider(selectedProvider, customerInfo)

// 5. Show support information provided by service providers to customers

displayResponse(response)

Requirement C: Compliance assessment service providers can obtain customers in the system and provide compliance certification services for customers.

// Define a compliance assessment service provider class

public class ComplianceProvider {

    // Define a method for obtaining customer information

    public Map<String, String> getCustomerInfo() {

        // Obtain customer information from the system

        Map<String, String> customerInfo = System.getCustomerInfo();

        return customerInfo;}

    // Define a methodology for conducting compliance assessments

    public String evaluateCompliance(Map<String, String> customerInfo) {

        // Conduct compliance assessment of customer information

        // Returns the result of the compliance evaluation

        return complianceResult;}}

// Implement the following steps in the system:

// Create a compliance assessment service provider object

ComplianceProvider complianceProvider = new ComplianceProvider();

// Obtain customer information

Map<String, String> customerInfo = complianceProvider.getCustomerInfo();

// Conduct compliance assessment of customer information

String complianceResult = complianceProvider.evaluateCompliance(customerInfo);

  • In sections 4.1.1 and 4.1.2, we focused on introducing the implementation of the system from the perspective of logical view and process view to make it easier for readers to understand, and therefore did not cover the details of specific modules' technical implementation. As for computational resources, we discussed it at the beginning of 4.1 System Architecture Design:

During the process of architecture design, it is difficult to estimate the traffic load of the system due to the new business scenario. If the traditional approach of deploying applications directly on cloud servers is adopted, it would require manual horizontal scaling when faced with insufficient traffic to support current circumstances. Pre-provisioning multiple servers in advance could lead to resource waste. Therefore, based on business requirements, a Serverless architecture based on AWS was adopted for implementation. By deploying microservices to AWS Lambda and pushing business tasks to the network edge, the application can automatically scale elastically according to traffic volume, and the cost can be calculated based on the number of requests and computing time.

  • In section 5, we added the "Experimental Environment" section to explain the computational resources needed to run the system. Based on the online performance test, the system only requires a monthly operating cost of $20, while renting a cloud server with four cores and 16GB memory costs $70 per month. Additionally, Serverless architecture eliminates the need for maintenance and management, resulting in a reduced operational cost. These results indicate that adopting a Serverless architecture can significantly lower the system's overall cost.

Comment 2: The discussion of the results, as well as the analysis of the experiments, are not very clear. Because you supplied in Figure 12 the entire scenarios involve in your work, there is also a need to provide each experimental scenario separately along with its own outcome separately.

Response 2: Thank you for your suggestion. Your advice is very helpful in improving the quality of our paper. We have restructured the "Results and Discussion" section and added an "Experimental Environment" subsection to provide a detailed description of the experimental setup. The specific content is as follows:

The main objective of this section is to validate the advantage of automatic scaling in Serverless architecture by conducting load and stress testing on the system. To avoid a single server's network throughput from being too high, multiple servers are introduced to simultaneously send requests. The system's load capacity is described based on several dimensions, including average response time, total number of requests, successful and failed requests, and throughput rate (RPS). In addition to verifying the stability and reliability of the program, stress testing can also evaluate the application's load capacity based on test results, providing a reference for further program optimization.

5.1 Experimental Environment

Taurus is a tool for running various open-source load and functional tests. The experiment set up two Ubuntu 20.04 systems. One of the devices used the Taurus testing tool to test the platform designed in this study. For the load testing tool Taurus, a pre-packaged distributed load testing solution is available, which encapsulates Taurus as a container and can be easily deployed to multiple test servers. Test parameters such as the number of test servers, threads, target APIs, duration, etc. can be set through the front-end page. The entire testing process is shown in Figure 14.

5.2 Test Methodology and Results

The testing in this study was conducted using two methods: load testing and stress testing.

Comment 3: The explanation of how Figure 13 is interpreted is not very good, and Figure 13 itself is not very clear.

Response 3: Thank you very much for your suggestion. It has been very helpful in improving the quality of our article. We have provided a detailed description of the test data involved in Figure 13(new version Figure 15) and have processed and replaced the image accordingly. The specific changes are as follows:

The testing in this study was conducted using two methods: load testing and stress testing.

The load testing method involved using Taurus to interact with the system's API interface. By simulating user requests, the API return results were obtained in what is known as a single request. The server was subjected to stress load testing by increasing the number of servers and Taurus threads. Figure 15 shows the distributed test output using Taurus with 100 servers and 100 threads per server. In the scenario of a large number of distributed request loads within a short period, the self-scaling system proposed in this study expanded the bandwidth resources to 1.98 Kps, with an average latency of 1.82070s and a system response error rate of 0.0098%. The comprehensive test results indicate that the system we designed is capable of maintaining high performance, reliability, stability, and information processing capability when facing a large number of requests.

Figure 15. Taurus Test Outputs

Comment 4: There is a significant lack of depth in the description of the outcome in Figure 14, and there is a pressing need to bring attention to the specifics of those descriptions.

Response 4: Thank you for your suggestion. As mentioned above, we have re-analyzed and described the test data involved in Figure 14( new version Figure 16) in detail. The specific content is as follows:

During stress testing, a large number of concurrent users or requests are simulated using tools to evaluate the system's ability to handle a heavy load and maintain key performance metrics, such as response time, availability, and scalability during peak loads. As shown in Table 4 the testing results reveal the system's response time and success rate metrics when the number of testing servers ranges from 1 to 50. When 100 servers are deployed, each with 100 threads, the system's peak RPS is achieved, processing 5425 requests per second. However, the average response time of requests is around 1.8 seconds, attributed to network congestion caused by the excessive number of Taurus threads simultaneously sending requests from a single testing server.

Through CloudWatch monitoring, the growth curves of the backend Lambda invocation and instance scaling are shown in Figure 16. As the number of requests increases, Lambda automatically scales its service instances to handle the large volume of requests. The system proposed in this paper performs self-scaling of service resources based on actual request traffic and environmental demands, avoiding resource shortages or waste caused by rigid, centralized software architectures.

Table 4. Load Test.

Number of Servers

Number of Concurrent Threads

Throughput

Average Response Time

Total Number of Requests

Number of Failed Requests

1

100

49.8/s

1.97874 s

29830

0

5

100

268.52/s

1.8458 s

160843

0

10

100

549.16/s

1.79907 s

329493

0

50

100

2698.09/s

1.82738 s

1618854

0

100

100

5425.95/s

1.82138 s

3250144

3

Figure 16. Invocations and Concurrent.

In this section, the advantages of combining Serverless architecture and microservice framework with automatic scaling were verified through load testing and stress testing. The test results showed that the system was able to expand computing resources based on the number of requests without any operational intervention, and the throughput was able to meet the needs of most businesses. When faced with attacks such as denial of service due to a large number of requests, the platform we designed is also able to avoid single points of failure, meeting the requirements for stability and security of the system.

According to your comments, we have made extensive modifications to our manuscript to make our results convincing. Thank you again for your positive comments and valuable suggestions to improve the quality of our manuscript.

Round 2

Reviewer 1 Report

The updated paper can be accepted for publication.

Some minor English edits can be considered.

Author Response

We sincerely apologize for any inconvenience caused by the quality of our English writing. To address this, we enlisted the assistance of native English speakers to annotate and revise our entire manuscript.

Reviewer 2 Report

The authors have not addressd some of the previous comments. The following comments must be addressed before considering this paper:

1. Motivations of the paper are not clear.

2. The mentioned point-wise contributions are not precise. The structure of the Introduction section can be improved.

3. Related schemes are not discussed properly. The authors are suggested to include the following recent references:

a) Internet of Things-based healthcare system on patient demographic data in Health 4.0

b) Block switching: A stochastic approach for deep learning security

c) User privacy prevention model using supervised federated learning-based block chain approach for internet of Medical Things

d) Spam detection using bidirectional transformers and machine learning classifier algorithms

4. Algorithm is not mentioned.

5. The proposed scheme is unstructured. It is hard to identify the novelty of the proposed work.

6. Equations and figures are not represented properly.

7. Technical discussion on results is not given. Moreover, the results are not convincing.

8. The English language is very poor.

9. The organization of the paper is poor.

10. Important references are missing and all the details of the references are not given.

The English language must be improved.

Author Response

Comment 1: Motivations of the paper are not clear.

Response 1: We apologize for any lack of clarity in our previous statement. In the latest version, we have carefully revised the Motivation section. In this section, we emphasize the underlying motivation for our research:

This paper endeavors to investigate the design framework of a cross-border logistics compliance platform founded on the principles of microservice and serverless architecture. Specifically, the study aims to explore the amalgamation of microservice and serverless architecture in augmenting the security of cross-border logistics compliance transaction platforms, while encompassing the platform's design, development, and testing phases. The research undertaken in this paper will contribute to enhancing the stability and security of IoT applications, thereby serving as a valuable reference for related studies.

Comment 2: The mentioned point-wise contributions are not precise. The structure of the Introduction section can be improved.

Response 2: Thank you very much for your suggestion. We have refined the contributions based on your feedback. The specific contributions are as follows:

  • This paper proposes a comprehensive system design that integrates the Serverless and Microservice architecture paradigms, with a specific focus on the context of cross-border logistics. The developed system provides robust evidence of its practical advantages;
  • This paper presents the specific design of an API gateway and asymmetric encryption method for a cross-border logistics compliance platform, aiming to enhance security of the platform;
  • The study evaluates the architectural design pattern of Serverless and Microservice, analyzing its benefits in terms of system security, resource utilization, throughput, and latency.

We have made concise revisions and structural adjustments to the Introduction section to present the content more clearly. The current structure of the Introduction section is as follows:

  1. Background and System Requirements: We initially introduce the background and system requirements of cross-border logistics, providing an overview of the significance and challenges in this field.

  1. Motivation: In 1.1 Motivation section, we elaborate on the motivation behind our research, explaining why we chose to explore and apply a security solution that combines microservice and Serverless architecture.

  1. Research Challenges: In 1.2 Research Challenge section, we briefly discuss the technical challenges related to the research motivation, including challenges in service decomposition and application programming interface design.

  1. Contributions: Lastly, in 1.3 Contributions section, we summarize the findings and major contributions of our research, providing an overview of the effectiveness and advantages of the proposed architecture.

Through this structure, we are able to clearly introduce the background and requirements of the research, elucidate the research motivation, discuss the relevant technical challenges, and accurately summarize the results and contributions of the study.

Comment 3: Related schemes are not discussed properly. The authors are suggested to include the following recent references:

  1. a) Internet of Things-based healthcare system on patient demographic data in Health 4.0
  2. b) Block switching: A stochastic approach for deep learning security
  3. c) User privacy prevention model using supervised federated learning-based block chain approach for internet of Medical Things
  4. d) Spam detection using bidirectional transformers and machine learning classifier algorithms

Response 3: Thank you for your suggestion. We have added two references [7] and [8] to the related work section. Furthermore, the two additional references are not highly relevant to the content of this section; therefore, we did not include them in this section. Additionally, considering that our research encompasses aspects such as system security, resource allocation, and latency, we have provided introductions to three specific areas: IoT security, network resource allocation, and distributed architecture.

  • Internet of Things-based healthcare system on patient demographic data in Health 4.0
  • User privacy prevention model using supervised federated learning-based block chain approach for internet of Medical Things.

Comment 4: Algorithm is not mentioned.

Response 4: Since this paper focuses on the design of a platform for cross-border logistics, with a particular emphasis on the architecture and the design of various modules, algorithms are not the main focus. Therefore, there is limited content in the paper specifically dedicated to explaining algorithms.

Comment 5: The proposed scheme is unstructured. It is hard to identify the novelty of the proposed work.

Response 5: Thank you very much for the review and feedback on our paper. In response to your comments regarding the lack of structure in our approach and the difficulty in determining the novelty of our work, we have made further revisions and improvements.

To address the issue of non-structured approach, we have summarized and organized the system design section to provide a clearer and more systematic description of our approach, ensuring that readers can understand our work more effectively. The specific modifications are as follows:

We have adopted the process of domain-driven design, which involves clarifying the business context and user vision to provide direction and guidance for the entire system design process. This helps identify the problem domain and establish a common language for better communication and understanding among the design team members, reducing misunderstandings and errors. Additionally, techniques such as user story mapping, event storming, and command storming have been employed to facilitate a better understanding of user requirements and system functionality, ensuring that the system meets user expectations. Lastly, identifying aggregates and defining bounded contexts helps the design team better define the boundaries and components of the system, avoiding overly complex or simplistic system designs.

Regarding the issue of novelty, we have refined and modified the Contribution section in 1.3 based on Comment 2 to emphasize the research value and innovative aspects of our work in the field.

Thank you for your valuable feedback, and we believe that these revisions have addressed the concerns raised.

Comment 6: Equations and figures are not represented properly.

Response 6: Thank you for your feedback on the representation of equations and figures in the paper. We apologize for any confusion caused by their improper presentation. To address this concern, we ensure that all figures in the revised version of the paper are properly represented. We carefully review the formatting and layout to ensure clarity and readability. As for equations, Due to the focus of our paper on system platform design, there is limited mention of formulas in the text. Additionally, we provide detailed captions and labels for the figures, and ensure that figures are appropriately formatted and referenced within the text.Some details about the figures are as follows:

  • To develop this system, we conducted in-depth research and discussions with domain experts on the customers, compliance evaluation service providers, and platform administrators who participate in cross-border logistics business, and obtained the product vision board as shown in Figure 1.
  • User story mapping can assist in organizing user stories into useful models to understand system functionality, identify gaps and omissions in the backlog, and effectively plan overall releases that provide value to both users and businesses.
  • The aggregates are classified by contextual meaning. The final bounded context result is shown in Figure 3.
  • The overall platform design of the system is shown in Figure 4.
  • The logical view of the cross-border logistics system platform is shown in Figure 5. The figure shows the logic structure of the designed system.
  • As shown in Figure 6, the process view of the system is illustrated by a typical scenario where users send requests.
  • The current system employs a front-end/back-end separation design, leading to separate deployment pipelines for both. The following will introduce the design of the front-end and back-end, communication pattern of microservices, and the deployment design of persistence. The continuous deployment pipeline for the frontend is shown in Figure 7. The backend deployment pipeline is illustrated in Figure 8.
  • As illustrated in Figure 9, in a traditional application architecture, clients usually access the application’s APIs directly.
  • To address the aforementioned issues, the system design employs an API gateway to encapsulate the microservice APIs [52]. The gateway intercepts all incoming request data and forwards the requests to backend services. The architecture of the API gateway is illustrated in Figure 10.
  • By using Amazon SNS and Amazon SQS together, a message can be delivered to multiple consumers at the same time. Figure 11 shows the integration design of Amazon SNS and Amazon SQS.
  • Figure 12 and Figure 13 show the process of encrypt and decrypt.
  • To clearly describe the experimental environment, we take Figure 14 to show the entire testing process.In addition, the results are shown in Figure 15 and Figure 16.

We appreciate your valuable input, and have made every effort to improve the presentation of figures in the revised version of the paper.

Comment 7: Technical discussion on results is not given. Moreover, the results are not convincing.

Response 7: Thank you for reviewing our research and providing valuable feedback. We have carefully read your comments and understand your concerns. We would like to address the following points you mentioned:

Technical discussion: We acknowledge that in our paper, we may have focused too much on the implementation of system functionality and fulfilling business requirements, without providing sufficient elaboration on the technical discussion of the results. We will carefully examine our research data, conduct additional experiments if necessary, and provide a robust analysis to strengthen the validity and reliability of our findings.

Lack of convincing results: The experimental results presented in our study are derived directly from the output of the testing tools. Additionally, the data tables provided are meticulously obtained through the experimental process. Therefore, we believe that the experimental results are compelling and can be trusted. We appreciate your valuable input and assure you that we will make the necessary improvements to address the lack of technical discussion and enhance the persuasiveness of our results in the future.

Comment 8: The English language is very poor.

Response 8: We sincerely apologize for any inconvenience caused by the quality of our English writing. To address this, we enlisted the assistance of native English speakers to annotate and revise our entire manuscript.

Comment 9: The organization of the paper is poor.

Response 9: Thank you for your suggestion. We have refined the content of the first part, Introduction. Additionally, upon reviewing the entire paper, it is organized into six parts.

The first part provides a detailed explanation of the research motivation and challenges.

The second part describes and summarizes the relevant work in the field.

In the third part, we apply the domain-driven design method to analyze and divide the microservices involved in cross-border logistics.

Building upon the microservice division design in the third part, the fourth part focuses on describing key aspects of the platform, including its overall front-end and back-end architecture, microservice communication methods, and system storage methods.

Using the design from the fourth part as a foundation, the fifth part discusses the construction of the platform and presents system testing results conducted with testing tools.

Finally, the last part provides a summary of the entire research paper.

Comment 10: Important references are missing and all the details of the references are not given.

Response 10: We greatly appreciate your feedback regarding the missing important references in our paper. We have carefully reviewed our reference list and identified several crucial references that were indeed omitted. In the revised version, we ensure that our paper contains all the necessary citations.

Furthermore, due to space limitations, we only provided a description of the approaches in the referenced papers, without delving into the detailed technical aspects. In the future, we may consider organizing and presenting the technical details of the relevant works in a separate comprehensive article or review, allowing for a more thorough examination of the literature.